# A principle-based framework to determine countries' fair warming contributions to the Paris Agreement

Mingyu Li [1,2], Setu Pelz [3], Robin Lamboll [2], Can Wang [1] & Joeri Rogelj [2,3,4] ✉

Equity is a cornerstone of global climate policy, yet differing perspectives mean that international agreement on how to allocate mitigation efforts remains elusive. A rich literature informs this question, but a gap remains in approaches that appropriately consider non-$CO_2$ emissions and their warming contributions. In this study, we address this gap and define a global warming budget applicable to all anthropogenic greenhouse gases that is allocated to countries based on principles drawn from international treaties and environmental law. We find that by 2021 a range of 84 to 90 countries, including but not limited to all major developed countries, exhausted their budget share compatible with keeping warming to 1.5 °C (with 50% likelihood) under all allocation approaches considered in this study. A similar picture emerges for limiting warming to 2 °C (with 67% likelihood). A large group of countries will hence exceed their fair shares even if their pledges under the Paris Agreement represent their deepest possible emission reductions. Considerations of fairness should therefore start exploring aspects beyond domestic emissions reductions.

International climate negotiations have long debated fairness considerations to distribute efforts towards achieving common global targets. Dating back to 1992, the United Nations Framework Convention on Climate Change (UNFCCC)[1] states that all countries should prevent dangerous interference with the climate system in accordance with their "Common but Differentiated Responsibilities and Respective Capabilities" (CBDR&RC). The Paris Agreement reiterates this position and relies on countries voluntarily determining their contributions "on the basis of equity, and in the context of sustainable development and efforts to eradicate poverty"[2], while also qualifying CBDR&RC with the consideration of different national circumstances. Nearly all governments have submitted Nationally Determined Contributions (NDCs) under the Paris Agreement[2] to contribute to the agreement's aims to hold warming well-below 2 °C while pursuing to limit it to 1.5 °C[3], and around one-third have submitted long-term

strategies[4]. However, the broad coverage of NDCs does not guarantee that normative concerns are met[5] as countries seldom account for their full historical responsibility for causing climate change, or their relative capability to meet deep emissions reductions[5]. Achieving appropriate mitigation action globally requires equity and fairness considerations to be a central part of the solution, both to ensure access to sustainable development and to encourage countries with greater responsibilities and capabilities to take up appropriately ambitious climate actions[6].

The Paris Agreement leaves space for Parties to interpret the principles contained therein and to define how their NDCs are both fair and ambitious. Numerous allocation approaches[7–11] have been suggested to quantify fair shares under 1.5 °C or 2 °C-consistent pathways. They commonly consider factors including historical emissions[7] and indicators of socioeconomic development (GDP[8],

[1]School of Environment, Tsinghua University, Beijing, China. [2]Centre for Environmental Policy, Imperial College London, London, UK. [3]Energy, Climate and Environment Program, International Institute for Applied Systems Analysis, Laxenburg, Austria. [4]Grantham Institute—Climate Change and Environment, Imperial College London, London, UK. ✉e-mail: j.rogelj@imperial.ac.uk

income[9], population[10], energy production[11], etc). Other studies[12–14] introduced alternative approaches including composite indicators. These attempts claimed to bridge different perspectives, but have been criticized for combining contradictory principles and being logically inconsistent (e.g. decent minimum standards of living are combined with grandfathering[15]). Despite agreement on the general principles, it is arguably impossible to reach a political consensus on a single equitable effort-sharing among governments globally. Nevertheless, quantifications of fair national budgets hold important implications for informing the ethical obligations of countries. For example, over-emitting countries can be considered disproportionately responsible for climate-related loss and damages and should own corresponding compensation or reparation obligations[16]. Thus, fair shares serve as evidence informing both domestic and international ambition and action[17,18].

The Paris Agreement global temperature goal implies a limit to the total amount of $CO_2$ emissions that can ever be emitted[19], known as the total carbon budget. Several effort-sharing studies focus on allocating this total budget instead of a specific global emission pathway[20,21], which enables governments to determine their individual pace of reduction. However, these studies typically focus solely on $CO_2$ emissions[22]. A recent expansion of the use of the Global Warming Potential metric (known as GWP*)[23,24] helps to convert non-$CO_2$ greenhouse gases (GHGs) like methane ($CH_4$) to $CO_2$ warming-equivalent ($CO_2$-we) emissions and enables them to be incorporated in budget-based allocation procedures. Using GWP*, however, requires normative decisions on the period over which emissions are considered as this choice materially shifts relative warming contributions[25].

In this study, we present a global-warming allocation method for keeping warming to 1.5 °C or 2 °C, based on three interpretations of equity principles aligned with international environmental law. We then allocate the total global warming between 1850 and 2050 to individual countries and derive their remaining $CO_2$-we budget after 2021. We assess factors that have the largest effect on national shares, including normative considerations, and explore several methodological uncertainties. The calculated emission allocations would rectify the historical unfairness of warming contributions while reaching net zero $CO_2$-we emissions in 2050. By comparing the fair allocations with the deepest possible national emissions reductions, we open a discussion on the gap between ethical norms and real-world implementation, and explore potential remedies.

## Results

### Interpretation of principles for allocating fair shares

Allocating national warming budgets fundamentally relies on interpreting principles which underpin international consensus on climate action as captured by the Paris Agreement, such as equity and CBDR&RC[2]. Alongside and aiding their interpretation, we also consider the principles of harm prevention and precaution, described in international environmental law, motivating the avoidance of harm to other nations due to domestic actions and the pursual of efforts to limit climate change in spite of scientific uncertainties[26]. We consider these alongside related principles for equitable effort sharing discussed in the climate equity literature[15]. This includes the ability-to-pay, beneficiary-pays, and polluter-pays principles, which we consider in addition to the overarching principle of equality. In total, we develop three interpretations of how these principles can be

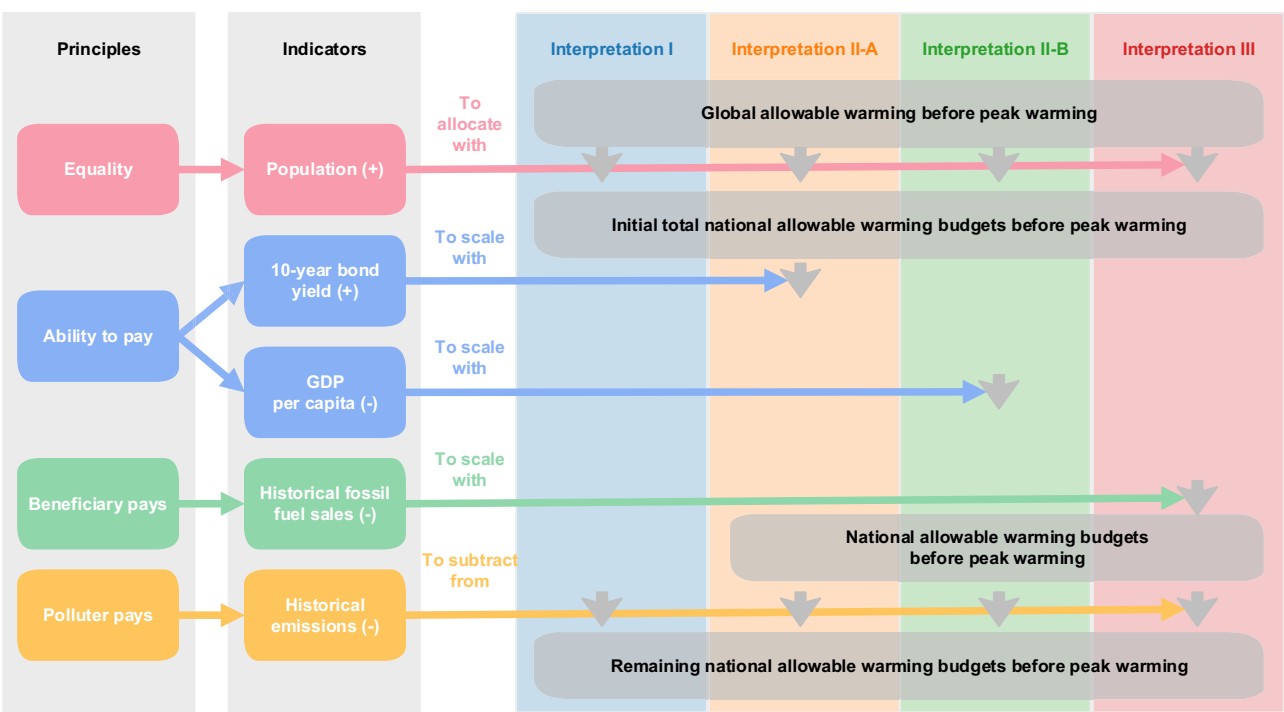

Fig. 1 | Analysis framework linking principles, indicators, and the allocation procedure to estimate fair global warming contributions. Each principle and the corresponding indicators are marked in different colours: pink stands for the principle of equality, light blue for ability to pay, light green for beneficiary pays and yellow for polluter pays. Interpretations are marked in different colours: dark blue stands for Interpretation I, orange for Interpretation II-A, dark green for Interpretation II-B and red for Interpretation III. The horizontal coloured arrows illustrate the incorporation of equity principles and indicators into the allocation procedure. Under the Indicators column, plus (or minus) signs in brackets indicate the indicators are directly (or inversely) proportional to the remaining national allowable warming budgets. Under the Interpretation columns, the vertical grey arrows suggest whether the corresponding indicator is included in this step.

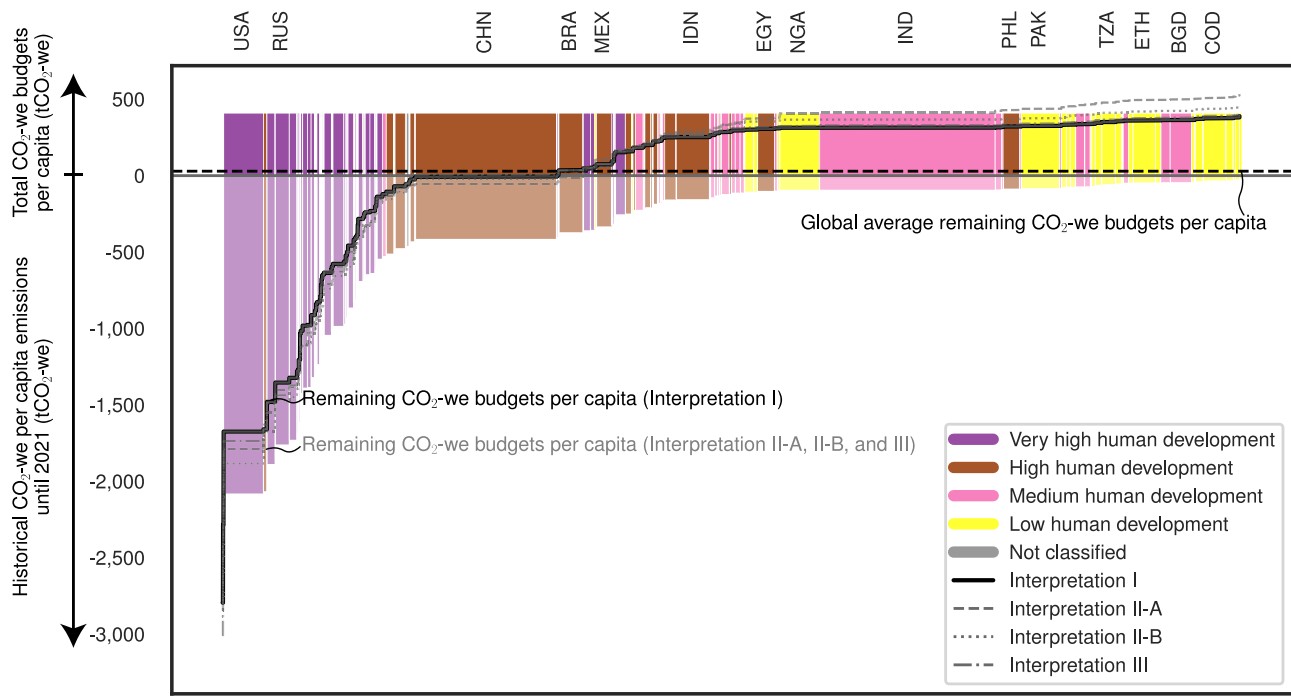

**Fig. 2 | Total (1850–2050), historical (1850–2021), and remaining (2022–2050) CO₂ warming equivalent budgets per capita by country.** Countries are ranked according to their remaining per capita national warming equivalent budgets (CO₂-we), under the central case which considers a historical warming contribution starting in 1850 and a global warming target of remaining below 1.5 °C with a 50% likelihood. The height of bars above the x-axis represents the total CO₂-we budgets per capita under Interpretation I that considers the equality and polluter-pays principles. The height of the bars below the x-axis represents the historical consumed CO₂-we budgets per capita. The width of the bars represents a country's population in 2050. The step lines represent the remaining CO₂-we budgets per capita per country with the black line referring to Interpretation I and the grey lines referring to Interpretation II-A, II-B, and III. Very high, high, medium, and low human development countries are marked in purple, brown, pink and yellow, and not classified countries in grey.

combined, to reflect the diverse moral perspectives frequently debated in climate negotiations (Fig. 1, and "Methods"). The grandfathering principle has been a longstanding concept in the literature, but it is not considered a basis for equitable distributions[15] and is therefore not included.

Our first interpretation (Interpretation I) relies on equality and the polluter-pays principle. Here we consider all living persons having an equal right to pollute, but inheriting country-based historic emissions extending back to two distinct starting years, 1850, and 1990. 1850 is chose as a common proxy year for the beginning of the industrial revolution and from which good data is available. Yet, some people may argue that society was unaware of the connection between CO₂ emissions and warming at that time, and ignorance may mitigate moral duties in some ethical frameworks. We therefore include 1990 as an alternative starting year. This is the publication year of the first assessment report of the Intergovernmental Panel on Climate Change (IPCC), marking a clear global consensus on the impact of the anthropogenic greenhouse effect. We determine the remaining warming from this start year to our target temperature, be it 1.5 °C or 2 °C, allocate this to individual countries, and subtract from this allocation countries' respective historic contributions to warming until 2021, thereby determining the remaining budget to net zero with due consideration of our interpretation of CBDR&RC.

Our second interpretation (Interpretation II) expands our first interpretation with consideration of the ability-to-pay principle, placing weight on different national circumstances. Ability to pay for mitigation is modelled as a function of the government bond yield (a proxy for government capacity to invest) (Interpretation II-A) and GDP per capita (Interpretation II-B). These indicators are used to scale the per-capita distribution of our first interpretation (Fig. 1).

Our third interpretation (Interpretation III) replaces the ability-to-pay with the beneficiary-pays principle, placing emphasis on the unequitable benefits that nations derived from their past emissions. We use total historical fossil fuel sales in present dollars attributable to the population in each country from the year 1900 to 2021 as a proxy indicator to reflect benefits, beyond historical contributions to warming. Also here, we scale the per-capita allocations from our first interpretation with this indicator before proceeding with the subtraction of historical contributions.

From a normative perspective, countries with higher GDP per capita, lower government bond yields, and higher benefits from historical fossil fuel sales should be allocated lower remaining global warming budgets. To avoid extreme punishment or encouragement of certain countries when mathematically scaling per-capita distributions, adjustments greater than two standard-deviations from the mean adjustment are capped to these limits, reflecting a value judgement taken in this work (see Methods for details).

**Fair national CO₂ warming equivalent budgets**

We explore the quantitative results through the lens of four country groups: very high, high, medium, and low human development countries, as defined by the United Nations Development Programme (UNDP) Human Development Index (HDI) ranking (Fig. 2, two alternative country categorization methods are provided in Supplementary Note 8, Supplementary Figs. 14–15). Of 68 very high human development countries, 59 are left with negative remaining budgets for the 2022–2050 period under equity Interpretation I, with historical warming contributions starting in 1850 and a global warming target of remaining below 1.5 °C with a 50% likelihood. Typical countries include the United States (US), the United Kingdom (UK), Russia, Japan, and

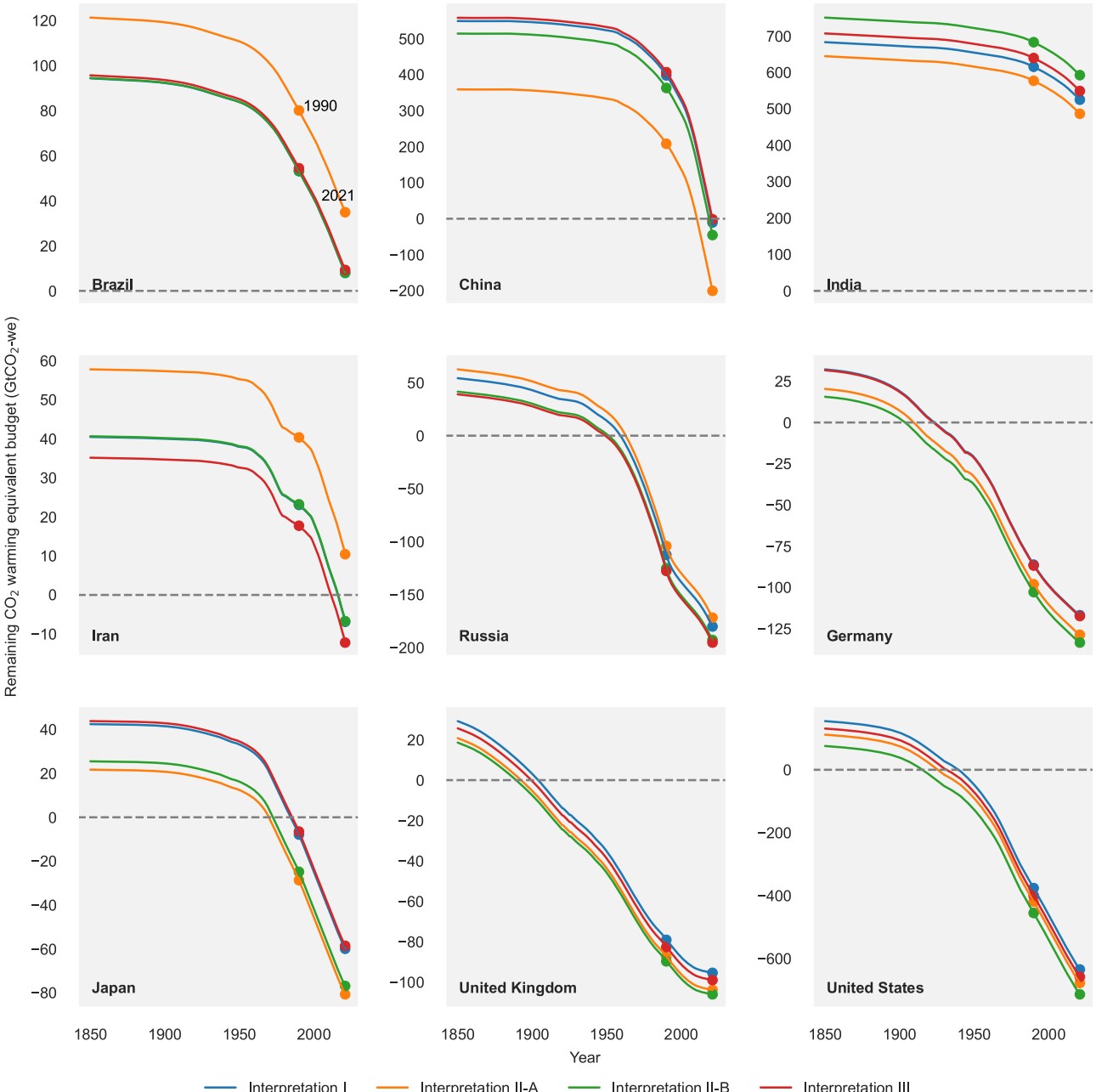

**Fig. 3 | The depletion of remaining CO₂ warming equivalent emission budgets over time.** The remaining budgets till 2050 for major countries over the period 1850–2021 are shown in coloured lines according to the different fair share interpretations: dark blue stands for Interpretation I, orange for Interpretation II-A, dark green for Interpretation II-B and red for Interpretation III. A country's remaining budgets until 2050 for a certain year represents the total budgets from 1850 to 2050 minus the historically emitted budgets before that year. Highlighted data points correspond to the years 1990 and 2021.

Qatar, due to their early industrialization and fossil fuel-based industries, or their economies based on oil and gas extraction. Notable exceptions with positive remaining budgets, such as Türkiye and Chile, are mostly developing countries. Over half of high human development countries (28 of 48) maintain positive budgets, including Indonesia, Brazil, Egypt, Philippines, and Mexico. These countries experienced later industrialization with lower historical emissions compared to very high human development countries. Countries like China, Iran, South Africa, Uzbekistan end up with a budget slightly below zero. Several high human development countries are among those with the lowest remaining CO₂-we emissions budget, including Palau, Ukraine, Bulgaria, Moldova, Bosnia and Herzegovina, and Turkmenistan. Most of these are countries from the former Soviet Union or transforming economies in Eastern Europe with high historical emissions. Palau, being a very small island nation, also has high per capita emissions. 37 of 42 medium human development countries have positive remaining CO₂-we budgets, including India, Bangladesh, Uganda, Kenya, and Iraq. All low human development countries have positive remaining CO₂-we budgets above the global average (29 tCO₂-we per capita). In total, globally, 86 countries had exhausted their 1.5 °C CO₂-we budget by 2021. This country group ranking remains robust against alternative equity interpretations explored in our Supplementary Note.

## Depletion of remaining budgets

Many very high human development countries have far exhausted their fair budget estimates even before 1990. Notably, the UK, France, Germany, Australia, and the US are among the first nations to exceed

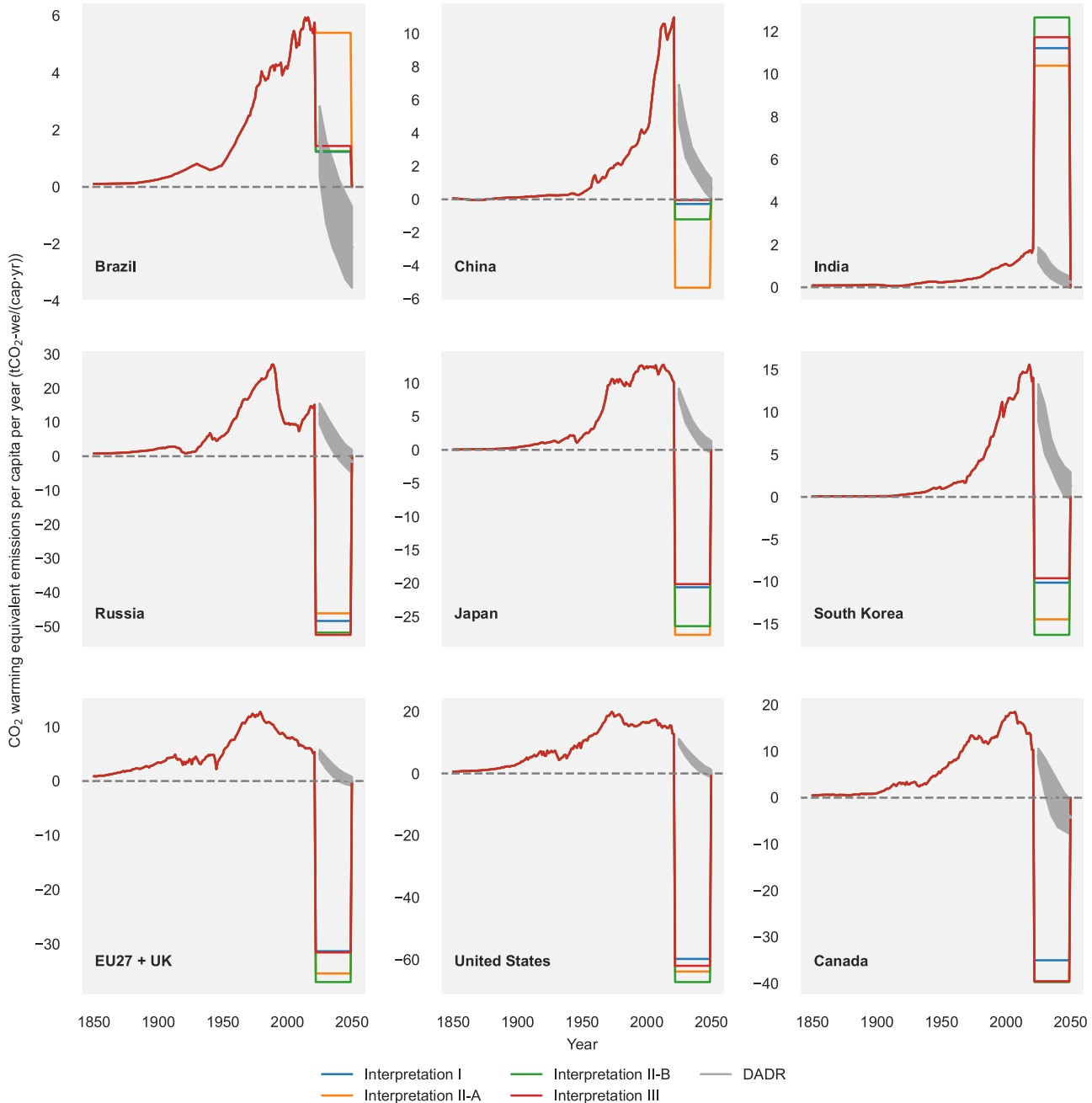

**Fig. 4 | The annual fair CO₂ warming equivalent emission budgets against the deepest available domestic reduction (DADR) pathways.** For the period 1850–2021, the historical CO₂-we emissions per capita data are shown in a single line for each country. For the period 2022–2050, the annual budgets per capita are shown in coloured lines according to the different fair share interpretations: dark blue stands for Interpretation I, orange for Interpretation II-A, dark green for Interpretation II-B and red for Interpretation III. For the period 2022–2050, shaded grey areas additionally show the DADR pathways from the IPCC Sixth Assessment Report database C1 category for each country or region, including CO₂ emissions only. The solid grey line within these areas indicates the median of the DADR pathways. EU27 + UK is short for the European Union (27 countries) and UK. EU27 + UK countries are treated as individual entities through the fair allocation process, and are combined only for illustrative purposes.

their budgetary limits (Fig. 3). South Korea still had 52% budget left in 1990 (compared to that in 1850) under equity Interpretation I, yet exhausted it rapidly by 2006. Despite the adoption of the UNFCCC in 1992, the slowdown rate is moderate. Russia is an exception with a notable turning point around 1990. However this is the result of the economic shock of the collapse of the Soviet Union, not of climate policy[27]. For high and medium human development countries, most had a large budget left in the 1990s. Yet, some soon exhausted it at an increasing rate after 2000. China's budget was exhausted in 2021. In 2021, India and Brazil still have 77%, and 8% of their estimated 1.5 °C

compatible budgets compared to the year 1850, respectively (see Supplementary Fig. 1a).

## Fair budgets against deepest available domestic reduction

Over the period 2022 to 2049, we consider the case where countries follow their fair allocations (Fig. 4). The framework here assumes that by 2050 unequal historical warming contributions are resolved and all countries return to zero CO₂-we emissions (Supplementary Figs. 1b and 17) by the time of peak warming. After 2050, the world is assumed to further maintain zero CO₂-we emissions, although long-term net-

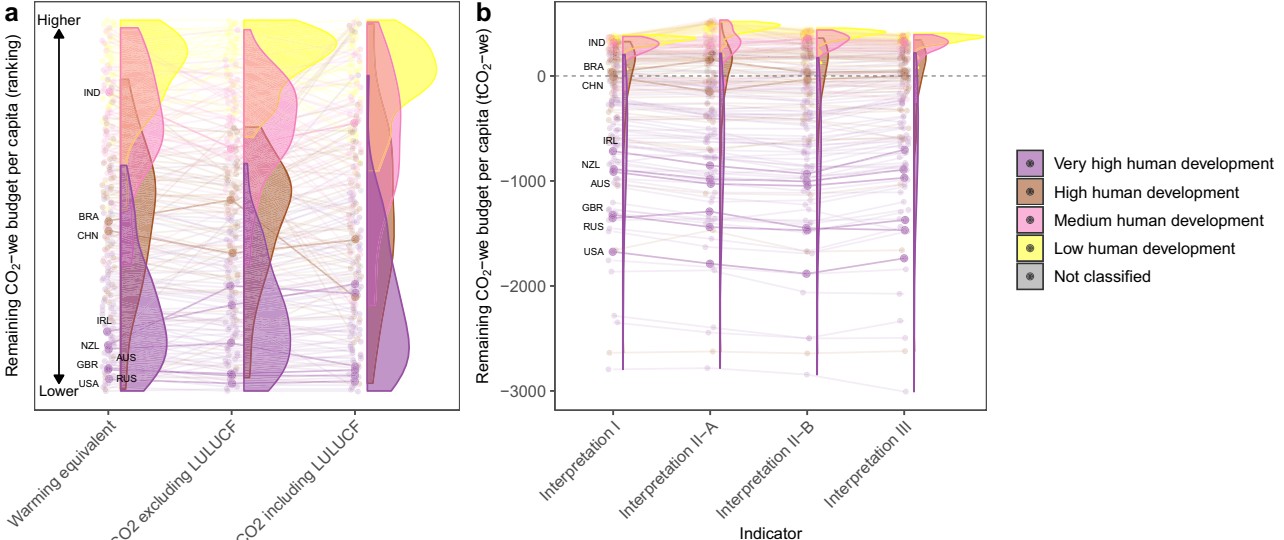

**Fig. 5 | The influence of accounting for different emissions sources and equity interpretations on national fair shares.** Influence of choice of greenhouse gas coverage (**a**), different equity interpretations (**b**) on remaining $CO_2$-we budgets per capita by country from 2021 onwards. **a** The x-axis shows different global allocation objects, comparing between allocations of global warming, $CO_2$ excluding land, land use change and forestry (LULUCF) and $CO_2$ including LULUCF. Both the global warming and $CO_2$ excluding LULUCF allocations exclude LULUCF emissions, while the $CO_2$ including LULUCF allocation accounts for historical LULUCF emissions. The y-axis shows ranking of national remaining $CO_2$-we budgets per capita, where higher budgets are positioned on the upper side and lower budgets on the lower side. The calculations in this panel apply Interpretation I. This comparison illustrates how the inclusion of gases in the allocation scope impacts country rankings. **b** The x-axis shows different fair share interpretations. In both figures, connected points represent data from the same country, showing how a countries allocation and relative position changes as greenhouse gas coverage or equity interpretations are changed. The overlapped density plots depict the distribution of remaining budgets by country group. Very high, high, medium, and low human development countries are marked in purple, brown, pink and yellow, and not classified countries in grey.

negative $CO_2$-we emissions and a gradual reversal of warming might be more aligned with the Paris Agreement[28] (Supplementary Note 1).

Our analysis estimates future national budgets consistent with specific fair share interpretations. Yet, these allocations may not be perceived as achievable domestically for certain countries. To explore this we take the set of emissions scenarios of the C1 category in the IPCC Sixth Assessment Report (AR6) scenario database that have country-level projections as a range of illustrative values for the deepest available domestic reduction (DADR) pathways, consistent with the target of limiting warming to 1.5 °C (>50%) with no or limited overshoot.

An important gap exists between many of the national fair share allocations and DADR pathways as estimated by global integrated assessment models (Fig. 4, Supplementary Fig. 16). The DADR pathways for the US, Canada, Russia and the European Union would result in about net-zero $CO_2$-we emissions by 2050, for China around 2060. However, even these deep reductions over the next decades are insufficient for these countries to stay within their fair remaining budget allocations as virtually all of them are negative. Brazil can roughly remain within its fair allocation due to its abundant carbon sink resources but there are large uncertainties surrounding the estimates of its deepest technically feasible emissions reductions. The remainder of India's equitable warming budget is about ten times the cumulative emissions under its DADR pathway. Indonesia, Pakistan, and Colombia are in a similar situation.

### Influence of normative considerations
Our analysis adopts three fair share interpretations (Fig. 1) that reflect different combinations of the principles drawn from treaties and international environmental law, and that are translated into quantified estimates using a set of key indicators. While Interpretation I serves as a central case for presenting our allocation results, it is not intended as a definitive solution but rather as an illustrative reference

point. Figure 5b shows how national remaining $CO_2$ budgets change with equity interpretations and the applied indicators.

In general, incorporation of the ability-to-pay (Interpretation II) or beneficiary-pays principles (Interpretation III) increases the divergence between country groups compared to a purely equality and polluter-pays approach (Interpretation I). Medium and low development countries get allocated more while very high human development countries less. This is expected when GDP per capita is taken as indicator for the ability-to-pay principle, as this indicator closely correlates with whether countries are categorized as developed, developing or least developed. Using purchasing power parity (PPP) or market exchange rates (MER) estimates of national GDP shows negligible differences (Supplementary Note 2, Supplementary Fig. 2). Using the 10-year bond yield as indicator for the ability-to-pay principle implies a greater emphasis on a government's credit and financial outlook. Consequently, emerging economies such as China, India, and Indonesia get smaller budget allocations, while countries like Russia, Iran, and Brazil get more given their lower economic expectations.

Using historical fossil-fuel sales as an indicator for the beneficiary-pays principle also does not dramatically influence the overall ranking of countries. Yet, the pattern is more complex compared to when GDP is used. Several countries across the various groups get markedly lower budgets than under other interpretations, resulting from a relatively higher reliance on fossil fuel production during their capital accumulation process. Notable examples include the UK, USA, Norway, Canada, Australia, Trinidad and Tobago, Qatar, Bahrain, Kuwait, United Arab Emirates, and Saudi Arabia among very high human development countries, Libya among high human development countries, as well as Angola and Timor-Leste among medium human development countries. We also consider colonial histories of countries when calculating historical fossil-fuel sales by attributing the sales from a colonized country to the colonial power during the colonization

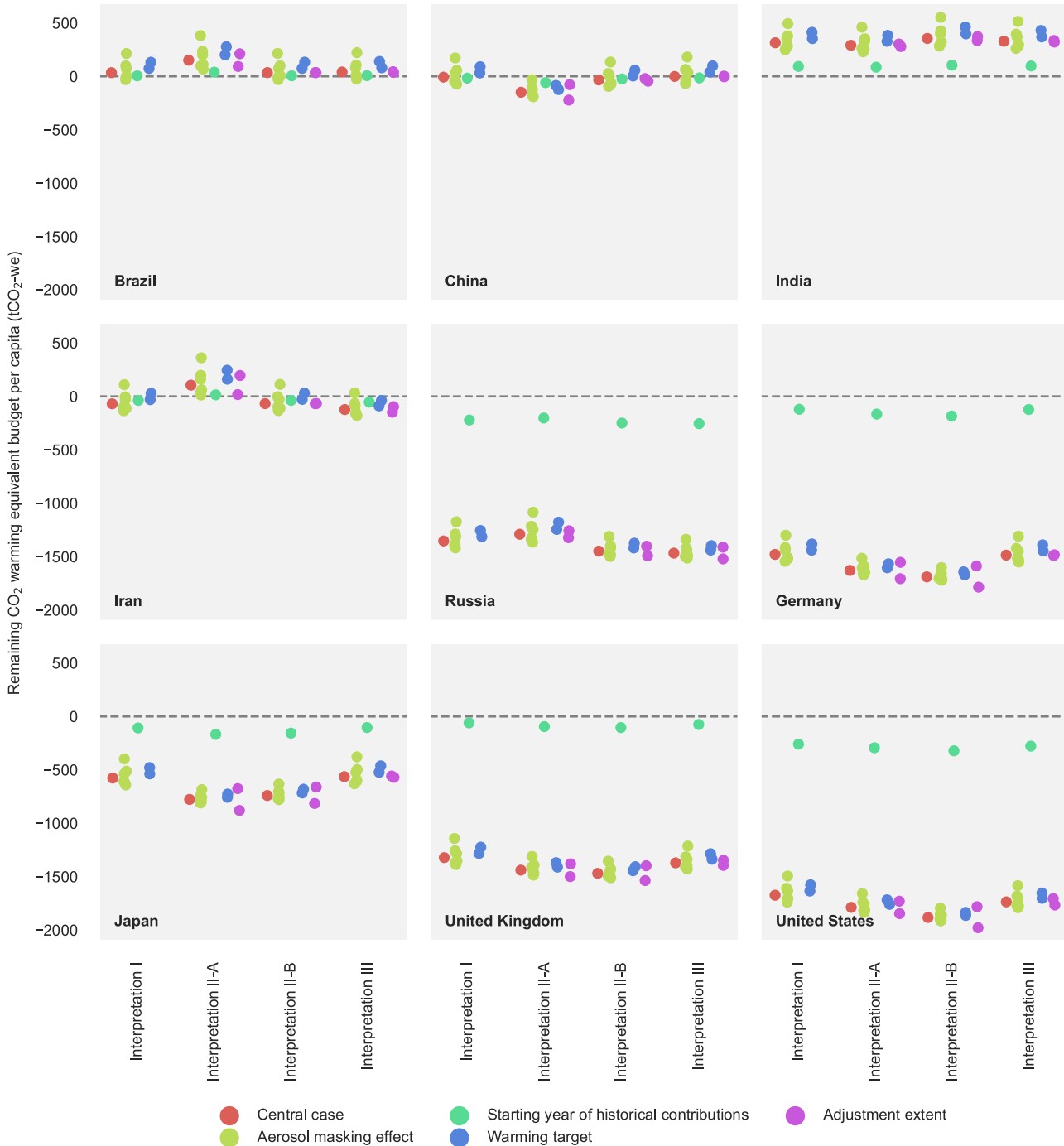

**Fig. 6 | Uncertainty in remaining (2022–2050) CO₂ warming equivalent budgets for major countries.** Each point represents an allocation case. For each country and each equity interpretation, the central case assumes a global warming target of remaining below 1.5 °C with a 50% likelihood, an indicator adjustment extent of 50% (the maximum adjustment when scaling per capita allocations), a historical warming contribution starting in 1850 and an aerosol masking effect at the 50$^{th}$ quantile. Alternative cases deviate from the central case by modifying only one parameter, with different parameter categories distinguished by colours.

Parameter variations are documented in Supplementary Table 3. Points of the same colour indicate multiple alternative choices within a specific parameter category: red for central case, cyan for starting year of historical contributions, purple for adjustment extent, green for aerosol masking effect, and blue for warming target. Adjustment extent refers to the degree to which a country's allocation is changed as a function of differences in driving indicators. Aerosol masking effect refers to the estimation of the aerosol masking effect at the time of peak warming. Warming target refers to the desired peak global warming target.

period to simulate the benefit transfer. The absolute influence of this additional operation on the remaining budgets is only minor (Supplementary Note 3, Supplementary Fig. 3).

Another normative choice we explore is the starting year from which countries' historical warming contribution is calculated, which is subject to debate. For example, 1850 is often taken as a proxy for the

start of the Industrial Revolution and the start of the increasing consumption of fossil fuels. Later years are also considered, such as 1990 (the release of the IPCC First Assessment Report) or 1992 (the establishment of the UNFCCC), arguing that excusable ignorance before the international treaty or recognized report may reduce the moral burden of the emitters. When 1990 is taken as starting year compared to 1850,

the difference in national remaining budgets decreases. This must be caveated with a recognition of the time-dependent warming effects of short-lived climate forcers as captured by the GWP* metric underpinning our analysis. Estimated contributions to warming by these gasses under this metric depend on changes in flux, such that later starting years of accounting may reward countries that subsequently reduce these fluxes, without considering their initial contribution to warming when fluxes were increasing. We find that under a 1990 accounting date, countries with positive budgets see a reduction and that countries with negative remaining budgets have less 'debt'[29]. However, the relative change tends to be smaller for high human development countries than for very high human development countries, indicating the relatively higher warming contributions of high human development countries after 1990 (Supplementary Note 4, Supplementary Figs. 4–5).

Various choices can affect budget outcomes in some countries, which can lead to gaming. When countries are asked to demonstrate the fairness and ambition of their long-term goals as mandated under the Paris Agreement, they may adopt choices that minimize their burden[5]. Typical very high human development countries with long fossil-fuel histories might favour later starting years for historical contributions or reject the notion of historical responsibility altogether. Very high human development countries like Switzerland and Finland, whose economies have historically and currently been primarily driven by high-value industrial sectors and the tertiary sector may emphasize that they have limited extraction of fossil energy and thus limited benefit from emissions. The Organization of Petroleum Exporting Countries may argue their economy is heavily reliant on fossil fuel production and therefore emphasise challenges associated with transitioning away from fossil fuel. Countries with high rates of poverty, like India, may underscore the development needs of their populations, emphasizing the necessity of allocation in proportion to population size. Within-country inequality is typically understudied across these distinct cases and appears a promising avenue to reconcile differing perspectives and starting points.

### Influence of uncertainty due to methodological factors and aerosol masking effect

In addition to the variation of estimates due to normative choices, uncertainties arising through model parameter settings and the quantification of the aerosol masking effects have also been explored. These include the degree to which a country's allocation is changed as a function of differences in driving indicators, the desired peak global warming target, and the estimation of the aerosol masking effect at the time of peak warming (Supplementary Table 3). The influence of these uncertain factors turns out to be less vital compared to the variations due to normative considerations (Fig. 6, Supplementary Note 5–7, Supplementary Figs. 6–13).

For instance, increasing the desired peak global warming target from 1.5 to 2 °C intuitively increases the remaining warming budgets for all countries. Countries whose remaining budgets shift from below zero under 1.5 °C to above zero under 2 °C, include South Africa, Israel, Iran, Cuba, North Korea, China, and Botswana. However, this adjustment's influence is less notable than the variability introduced by different estimates of the aerosol masking effect, comparable to the influence of adjustment extent, and barely noticeable compared to the substantial negative budget for major countries with very high human development. Additionally, countries' budgets show only minor changes in rankings. No country's budget shifts from below the global average to above it, and only Chile and Niue see their budgets move from above to below the global average. This suggests that qualitative implications of this study remain unchanged, with a clear distinction in remaining budgets among country groups (Supplementary Note 6, Supplementary Figs. 8–11).

## Discussion

Our analysis of fair warming shares shows a clear yet unsurprising distinction between very high, high, medium, and low human development countries. In addition, it also shows that under a range of normative considerations and scientific assumptions, most very high human development countries have exceeded their budgets already by the 1990s. Critical disparities exist between fair budget allocations and DADR pathways, indicating the need for additional measures beyond domestic emissions reductions to deliver an equitable contribution to the global mitigation challenge. The starting year of historical contributions is the most vital factor among those we examine, followed by the combination of principles and the choice of indicators to reflect them. Results in the main text focus on 1.5 °C-compatible budgets, but a similar picture emerges for 2 °C (see Supplementary Note 6).

By focussing on warming contributions, we manage to cover $CO_2$ as well as non-$CO_2$ gases, an aspect not addressed convincingly in previous allocation studies. The majority of the literature allocates fair shares excluding non-$CO_2$ emissions, which may overestimate remaining budgets for countries like Brazil, New Zealand and Ireland (Fig. 6a). The approach in this study ensures that historical warming contributions of countries with large non-$CO_2$ emissions are accurately accounted for, reconciling a key gap in the fair share literature. Substantial global non-$CO_2$ GHG reduction (for instance $CH_4$) offers the potential to limit peak warming and prevent surpassing critical climate thresholds[30]. If appropriately accounted for such reductions also help to limit the rapid depletion of fair shares[25,31]. Nevertheless, this work still does not resolve fair shares considerations for land, land uses change and forestry (LULUCF) emissions, which remain excluded due to scientific and normative issues. We do, however, show the implications if these are considered without resolving open questions (e.g. historical model uncertainty, warming due to carbon emissions from deforestation prior to 1850 and colonial or subsistence-based deforestation post 1850). If historical LULUCF emissions are included in the allocation scope, medium and high human development countries tend to be sensitive in the ranking of remaining per capita budgets, as LULUCF emissions constitute a notable portion of their historical $CO_2$ emissions (Fig. 6a).

Considerations of fairness in this study refer to global warming contributions only. Mitigation costs and climate change damages are not accounted for. A welfare perspective that includes these could lead to different allocations[32–34]. For instance, we assume that the impact of one unit of $CO_2$-we emissions remains the same regardless of when it is emitted. However, as climate damages increase nonlinearly with temperature rise, it has been argued that harsher economic penalties could apply for emissions occurring later[35].

This study also comes with several notes of caution in the interpretation of our findings. First, the chosen indicators should only be seen as proxies for the corresponding principles. For instance, fossil-fuel production is used to capture the benefit gained from GHG emissions. Yet, the evaluation of wealth can vary depending on the perspective of extraction and consumption, and the profits of a fossil fuel company may not necessarily align with the country where it extracts, nor have benefited the population as a whole. Second, the quantification of countries' historical contributions is based on production-based emissions within a national territory. Emission transfers through international trade add to the overall uncertainty. Considering value-chain or consumption-based emissions tends to increase developed countries' historical warming contributions[36], although also low-income countries without established industrial sectors are affected. Third, while this study considers all main GHG emissions, their mix varies across sectors. Consequently, mitigation potentials and costs can also vary markedly. The framework of this study could be extended to sectors instead of countries to understand fair sectoral contributions. Finally, several additional

equity perspectives are worth considering in future studies. The fundamental unit of importance in our allocation framework is an individual person in a country, which makes sense as nations are the primary actors in international laws and negotiations. Within countries, we consider all individuals to be equal. However, inequality exists among social groups. In-country equity concerns could be addressed, for example, by only counting emissions contributions from people above the poverty line[37], or considering basic living needs[38] as a threshold.

Many developed countries have already depleted their fair warming budgets across all fair share interpretations considered here. In addition, reaching net zero will still take time, adding further emissions to their tally. This fundamental exceedance of fair warming budgets needs to be acknowledged and can be reflected when setting targets. Countries that find themselves in this situation have a moral duty to consider implementing the deepest available emissions reductions domestically, combined with the scale-up of carbon dioxide removal (CDR)[39]. Permanent CDR measures serve as a potential way to pull down fair emission quota overshoot, as well as to address the uncertain climate response[40]. Ensuring an equitable deployment of CDR contributes to overall climate justice[41–44]. Yet, this requires careful management of potential side effects that can impact economic prosperity, well-being, environmental, and human health, with negative consequences possibly outweighing the positive[45]. Therefore, it is crucial that CDR is not overly relied upon at the cost of substituting short emission reductions or threatening other development goals[46]. Considering the unevenly distributed country-level CDR deployment potentials, overemitting countries may find themselves unable to compensate for their fair emission quota overshoot solely through domestic CDR[47]. These countries can assist mitigation efforts in other countries through financial support, technology transfer or capacity building, although estimating appropriate levels of support remains an area of active research[48]. Such international support, particularly climate finance, could also help developing countries to implement their conditional NDCs, further improving equity and enhancing ambition[49].

## Methods
### Calculation of allocation procedure
We allocate a global warming budget to 195 countries, covering all UNFCCC member states with the exception of the Holy See and the State of Palestine. The allocation procedure consists of four steps. Fair share Interpretations I, II-A, II-B, and III are formed through these steps. All interpretations include Steps 1, 2, and 4, while Interpretations II-A, II-B, and III also include Step 3.

**Step 1: determining the allocation target.** The first step is to determine the allocation target, which is the global warming budget from 1850 to 2050. Given that observed warming of surface temperatures is the coeffect of GHG warming and aerosol masking, we account for the cooling effect caused by aerosol masking. By removing the impact of aerosol masking from the global temperature target, we quantify the global warming budget of GHG contribution alone.

$$W^{1850-2050} = T^{1850-2050} - A^{1850-2050} \qquad (1)$$

where $T^{1850-2050}$ is the global warming target in 2050 compared to the 1850 level, $A^{1850-2050}$ the estimated global aerosol masking effect in 2050, with $W^{1850-2050}$ the global warming budget caused by GHGs in 2050.

**Step 2: allocating initial national allowable warming budgets.** In the second step, we interpret the equality principle to determine the initial national warming budget. We give each person living in a given year an equal budget for warming. Thus, the initial national warming budget of a certain country is derived by summing up the budgets of all

individuals within the country. The year 2050 is used as the default reference year for population data, to maintain the consistency of time with the allocation target.

$$w_i^{1850-2050, \text{ initial}} = \frac{W^{1850-2050}}{P^{2050}} \times p_i^{2050} \qquad (2)$$

where $w_i^{1850-2050, \text{ initial}}$ is the initial national warming budget of country $i$, $P^{2050}$ the global population in 2050, and $p_i^{2050}$ referring to the population of country $i$ where $i = 1,...,N$ ($N = 195$) worldwide countries.

**Step 3: adjusting national allowable warming budgets.** In the third step, we interpret either the ability-to-pay principle for Interpretation II or the beneficiary-pays principle for Interpretation III. We translate the principles with measurable indicators, including the 10-year bond yield (Interpretation II-A), GDP per capita (Interpretation II-B), and historical fossil fuel sales (Interpretation III). Since GDP per capita and historical fossil fuel sales are negatively correlated to the adjustment, we calculate adjustment coefficients by applying a reversed linear transformation and cap the indicators' extent to two standard deviations from the mean adjustment to avoid extreme mathematical outliers.

$$\alpha_{i,j} = \begin{cases} 1 - M\% & , & x_{i,j} \leq \text{mean}_j - 2 \times \sigma_j \\ 1 - M\% + \frac{x_{i,j} - (\text{mean}_j - 2 \times \sigma_j)}{2 \times \sigma_j} \times M\% & , & \text{mean}_j - 2 \times \sigma_j < x_{i,j} \leq \text{mean}_j + 2 \times \sigma_j \\ 1 + M\% & , & x_{i,j} > \text{mean}_j + 2 \times \sigma_j \end{cases} \qquad (3)$$

where $x_{i,j}$ is the value of indicator $j$ of country $i$ where $i = 1,..., N$ worldwide countries. $M\%$ is the maximum adjustment extent of each indicator. It is a universal factor, set so that $M$ times the number of indicators sums to a constant with a default value set as 50%. Alternative settings of 25% and 75% are also explored. $\text{mean}_j$ is the mean value of the data series for the indicator $j$, and $\sigma_j$ is the standard deviation value for the indicator $j$.

The larger the 10-year bond yield the smaller the intended adjustment. Thus, we implement a transformation following Eq. (4):

$$x_{i,10Y} = \frac{1}{1 + x_{i,10Y}^{\text{original}}} \qquad (4)$$

where $x_{i,10Y}$ is the 10-year bond yield value of country $i$ after transformation, and $x_{i,10Y}^{\text{original}}$ the original value.

The national warming budget is derived by applying the adjustment coefficients on the initial warming budgets and then normalizing them to the global total budget.

$$w_{i,j}^{1850-2050} = \frac{\alpha_{i,j} \times w_i^{1850-2050, \text{ initial}}}{\sum_{i=1}^{N}(\alpha_{i,j} \times w_i^{1850-2050, \text{ initial}})} \times W^{1850-2050} \qquad (5)$$

where $w_{i,j}^{1850-2050}$ is the national warming budget of country $i$ where $i = 1,...,N$ worldwide countries by the adjustment of indicator $j$.

**Step 4: calculating remaining national allowable warming budgets.** In the last step, we interpret the polluter-pays principle by accounting for historical contributions. The national remaining budget is derived by subtracting the historical contributions from the total national warming budget.

$$w_{i,j}^{2022-2050} = w_{i,j}^{1850-2050} - w_{i,j}^{1850-2021} \qquad (6)$$

where $w_{i,j}^{2022-2050}$ is the remaining national warming budget of country $i$ from 2022 to 2050 with the enforcement of indicator $j$, $w_{i,j}^{1850-2021}$ the historical consumed national warming budget of country $i$ from 1850 to 2021.

The warming budgets are converted into $CO_2$ equivalent emissions with the transient climate response to cumulative $CO_2$ emissions (TCRE).

$$e_{i,j}^{2022-2050} = \text{TCRE} \times w_{i,j}^{2022-2050} \tag{7}$$

where $e_{i,j}^{2022-2050}$ is the remaining national $CO_2$ equivalent emission budget. TCRE equals 0.45 °C/1000 GtCO$_2$ in this study, which is the best estimate according to the IPCC Sixth Assessment[50].

## Historical warming contribution

Historical emission data was obtained from PRIMAP[51], which is a comprehensive set of GHG emission pathways derived from several published datasets. The countries' historical GHG emissions are strongly dependent on the coverage of calculation, and it is necessary to specify the scope of coverage. In this study, the GHG gases include $CO_2$, $CH_4$, $N_2O$, and F-gases, and we use the national total that contains all sectors other than LULUCF and international aviation and shipping. The HISTCR data series from PRIMAP is used, where country-reported data are prioritized over third-party data.

The countries' historical warming is estimated to determine their utilized budget. To calculate the global contribution to the change in global mean surface temperature compared to 1850–1900 levels, we use a simple climate model (FaIR) to simulate the temperature response to emissions in the historic period, with the time series data input of globally emitted greenhouse gas and aerosol emissions[52,53]. We calculate a country's marginal contribution to global warming by removing a country's greenhouse gas emission series from the model input. Since the sum of the marginal contributions of each country is not necessarily equal to the global temperature rise, we normalize by equal proportions ensure the sum of contributions of each country add up to the global temperature rise.

$$w_i^{1850-2021} = \frac{w_i^{1850-2021,\,\text{marginal}}}{\sum_{i=1}^{N} w_i^{1850-2021,\,\text{marginal}}} \times W^{1850-2021} \tag{8}$$

where $w_i^{1850-2021}$ is the historical national warming contribution of country $i$ in 2021 compared to 1850–1900 levels after normalization, and $w_i^{1850-2021,\,\text{marginal}}$ the marginal warming contribution of country $i$ in 2021 compared to 1850–1900 levels. The global historical warming contribution contains both warming contribution from GHG gases and the aerosol masking effect. Under this calculation, the historical warming contribution of GHG gases equals 1.22 °C by 2021. The size of the calculated global $CO_2$ warming-equivalent emission budget from 2022 equals 282 GtCO$_2$-we for a median global warming limit of 1.5 °C and 1232 GtCO$_2$-we for 2 °C, consistent with the estimation of previous studies.

In the alternative case when the starting year of countries' historical warming contribution is taken to be 1990, we calculate a country's marginal contribution by excluding the emissions of that country from 1990. We then apply Eq. (8) and obtain the national warming contribution in 2021 compared to the 1990 level after normalization.

## Aerosol masking effect

The aerosol masking effect on current global warming is quantified by the climate model FaIR. We calculate the warming from each scenario in the AR6 scenario database[54] both with and without aerosol emissions. We then calculate the temperature change between 2019 and the year of peak warming in aerosol and non-aerosol scenarios and calculate quantiles of the difference between these two runs. We use the 2019 value as the estimate for the current aerosol masking effect. Across all scenarios, the mean of the medians for the 2019 aerosol masking value is approximately −0.4 °C, consistent with the IPCC AR6 Working Group 1 assessment[50]. This value indicates the extent to which aerosols are offsetting the warming effect of GHG emissions.

The aerosol masking effect in the target year is estimated with peak levels of global warming (Supplementary Table 4). We exclude scenarios that do not exhibit a peak temperature before 2100.

## Sensitivity analysis

We examine the sensitivity of fair national budgets by normative considerations as well as methodological and physical uncertain factors.

The normative considerations include different fair share interpretations and indicators, as well as the starting year of the historical warming contribution. To reflect the ability-to-pay principle, we also consider alternative indicators that can provide additional insights into a country's economic capacity, including GDP per capita measured in MER, GDP per capita measured in PPP and the 10-year government bond yield. The starting time of historical warming contribution is another parameter choice since the growth trajectory of emissions varies strongly from country to country. The default time is set to be 1850, which is often taken as a proxy for the start of the Industrial Revolution relative to which the goals of the Paris Agreement are expressed. We include 1990 as an alternative starting year, which is the release date of the IPCC First Assessment Report[55].

For methodological and physical uncertainty factors, we consider alternative parameter choices, as well as the uncertainty in geophysical factors (Supplementary Table 3). We explore the impact of adjusting the allocation scope when distributing the 1.5 °C target under the Paris Agreement. Specifically, we compare our allocations of the global warming budget with two alternative scenarios: (1) allocations of global $CO_2$ emissions excluding LULUCF, (2) allocations of global $CO_2$ emissions including LULUCF (Fig. 5a). Both scenarios follow the original allocation framework and use equity Interpretation I, while adjusting the allocation to the global $CO_2$ budget, which is set at a remaining 500 Gt, in line with the IPCC Sixth Assessment Report, which limits warming to 1.5 °C (50th percentile)[56]. The first scenario accounts for $CO_2$ emissions excluding LULUCF for historical emissions, that is encompassing all sectors except LULUCF and international aviation and shipping. The second scenario includes LULUCF in the $CO_2$ emissions. Historical LULUCF emission data is also sourced from PRIMAP. It is important to note that the allocation result generated when PRIMAP LULUCF data is included must treated with care as PRIMAP LULUCF data are constructed from different sources using different methodologies and are not harmonized.

## Calculation of annual historical depletion of $CO_2$-we budgets

We estimate a country's historical $CO_2$-we emissions from 1850 to 2021 and subtract them from its $CO_2$-we emissions budget to show the trend of its depletion over time. The allocation of cumulative $CO_2$-we emissions over time is proportional to a country's annual $CO_2$-we emissions.

$$e_{i,j}^{t} = \frac{c_i^{t}}{\sum_{y=1850}^{2021} c_i^{y}} \times e_{i,j}^{1850-2021} \tag{9}$$

where $e_{i,j}^{t}$ is the historical $CO_2$-we emissions of country $i$ under indicator $j$ in year $t$ where $t=1850,...,2021$, and $c_i^{y}$ is the $CO_2$ warming equivalent emission of country $i$ in year $y$. $e_{i,j}^{t}$ is subtracted from the total $CO_2$-we budget of a country to illustrate the allocated budget's depletion over time (Fig. 3).

The $CO_2$ warming-equivalent emissions capture the total warming effect of various GHGs, by equating the change rate of climate pollutant to $CO_2$ emissions. We calculate $CO_2$ warming-equivalent emissions for $N_2O$ and F-gases using GWP100. Considering $CH_4$ is a short-lived GHG, we calculate $CO_2$ warming-equivalent emissions for $CH_4$ using GWP*, which is a simple model to calculate warming-equivalent emissions[23].

$$c_i^{t,*} = g \frac{(1-s)H \Delta c_i^{t,100}}{\Delta t} + g s c_i^{t,100} \tag{10}$$

where $c_i^{t,*}$ are the $CO_2$ warming equivalent emissions of country $i$ in year $t$ using GWP*, and $c_i^{t,100}$ are the $CO_2$ warming equivalent emissions of country $i$ in year $t$ defined using GWP100 with a time-horizon $H = 100$ years. $\Delta c_i^{t,100} = c_i^{t,100} - c_i^{t-\Delta t,100}$, the change of $CH_4$ emissions during period $\Delta t$ as 20 years. $s$ is a coefficient defined in ref. 57, which equals 0.25. $g$ is a function of $s$, which equals 1.13 for $s = 0.25$ and $H = 100$ years.

## Data sources

The metrics used in this study can be categorized into three distinct time frames: historical data, current status quo data, and future data. The historical data refers to historical GHG emissions, aerosol emissions, and historical fossil fuel sales, which reflect a country's historical warming contribution or economic benefits from emitting. The current status quo data include 10-year bond yield, and GDP per capita, which demonstrate a country's ability to mobilize resources to combat climate change. The future data include the population and national DADR pathways from 2022 to 2050. For population data, we follow United Nations medium scenario projection[58], a scenario with medium fertility, medium mortality and medium international migration. The national DADR pathways from 2022 to 2050 are used as a comparison to our fair allocation result. They are taken from the IPCC AR6 database[54], following a target of limiting warming to 1.5 °C (>50%) with no or limited overshoot (C1 category). LULUCF emissions have been removed to maintain consistency in the comparison. Details can be found in Supplementary Note 9, Supplementary Tables 1–2.

In addition, we classify the countries into four country groups following the UNDP HDI tiers: very high, high, medium, and low human development[59]. Two alternative country categorization methods are included in the Supplementary Note 8. Colonial relationships between countries and their corresponding duration are derived from the Colonial Dates Dataset[60], which aggregates information of European colonial empires.

## Data availability

Data generated during this study have been deposited in the Zenodo database under accession code https://doi.org/10.5281/zenodo.14577948[61]. All raw data used in this study is publicly accessible via the cited literature.

## Code availability

The code used to produce the calculation results and figures in this study, have been deposited in the Zenodo database under accession code https://doi.org/10.5281/zenodo.14577949[61].

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

## Acknowledgements

This work was supported by the National Natural Science Foundation of China, grant no. T2261129475 and no. 72140002 (M.L. and C.W.), the China Scholarship Council programme, grant no. 202206210201 (M.L.), and the European Union's Horizon 2020 Research and Innovation Programme's ESM2025 project under grant number 101003536 (J.R.).

## Author contributions

J.R. initiated and supervised the research. M.L. led the analysis, created the figures and wrote the first draft. M.L., J.R. and S.P. jointly developed the framework and framing. M.L. implemented the warming equivalent budget code. R.L. estimated historical warming contributions of countries and the magnitude of the global warming effect. All authors including M.L., S.P., R.L., C.W. and J.R. contributed to writing, editing and revising the manuscript.

## Competing interests

The authors declare no competing interests.
