## [Peer Review file · Nature Communications]

A principle-based framework to determine countries' fair warming contributions to the Paris Agreement

Corresponding Author: Professor Joeri Rogelj

Version 0:

Reviewer comments:

Reviewer #2

(Remarks to the Author)

I find this to be a high-quality study touching on an important topic. The level of novelty for publication in Nature Communications is debatable as it is not exactly the first paper to look at fair shares of decarbonisation pathways. While the inclusion of non-CO₂ and LULUCF emissions does make it a more complete study than others, these additions do not really contribute an important advance that substantially changes the overall picture.

Nonetheless, I would overall support its publication (subject to revisions outlined below) as I find the study to be very effective at drawing attention to the equity considerations of global climate policy and would have broad interest.

General comments

The manuscript focuses on a 1.5°C budget with the 2.0°C budget relegated to the supplementary information with only a brief mention in the main text that a similar picture emerges for 2°C. I feel like the paper would be more impactful if this was flipped and it primarily focussed on 2.0°C. Firstly, many consider the 1.5°C target to be now be unrealistic so some readers might dismiss this analysis as only applying to an unrealistic scenario and overlook the important finding that the implications are essentially the same for 2°C. Secondly, the current state of NDCs and LTSs suggests around 2°C so focusing on 2°C would make the study more relevant to the current state of global climate pledges. This change would also emphasise the point that even though the Paris Agreement has been successful at producing pledges in line with the overall 2°C target there remain equity issues even if countries achieve their pledges (this aspect could also be elaborated on more in the Introduction).

For these reasons, I would find the article more interesting if it focused on 2°C but this is more of a personal opinion on what would make more impactful research rather than a flaw in the paper. In either case of the authors changing the focus to 2°C or staying with 1.5°C, the other budget needs far greater discussion in the main text other than just "a similar picture emerges for 2°C". It is not immediately obvious and intuitive the picture would be so similar considering that the remaining 2°C budget is roughly 5 times larger than the remaining 1.5°C budget.

It would also be useful to have some kind of summary table or figure to better illustrate the differences between the various scenarios and sensitivity tests. Figure 2 of the main text and Supplementary Figure 9 at first appear identical until spotting the big differences in the global average remaining budgets per capita. As a possible example, a table providing the average remaining budgets by country groupings might help with this. A similar issue arises with Supplementary Figure 7 as it is impossible to notice the difference this change in assumption has made to the results compared to Figure 2

The manuscript also does not sufficiently describe the categorisation of "developed" and "developing" countries and LDCs. The use of the M49 classification is not ideal as the distinction is not clearly defined (see "note on developed and developing regions" on the M49 webpage including in the authors' references). A better approach would be to classify the countries either by UNDP HDI tiers (very high, high, medium, and low) or World Bank country classifications by income (High, upper-middle, lower-middle, and low).

In my opinion, it would also benefit the results to have four groups instead of three (based on the HDI or income categories) as currently the "developing" countries group is very large and puts countries at very different stages of development

together.

The manuscript also does not state the overall size of the calculated remaining budgets. This should be mentioned somewhere so that the readers can see that it is consistent with other studies e.g. (Lamboll et al. 2023).

Methods

The population assumptions are not clear – presumably, the authors used the UN's medium scenario but this is not stated.

The projected population to 2050 is a key assumption for allocating the budgets so sensitivity analysis using the UN's high- and low-fertility scenarios should be provided.

It is not clear which countries have been included in the analysis – is this all parties to the UNFCCC? I do not think the overall N is stated in the manuscript and there is no explanation of which countries may have been included or excluded.

The EU27+UK countries appear together in Fig. 5 but seem to have been treated individually elsewhere as some countries are referred to individually. This should also be made clearer – is it a case of combining them just for illustrative purposes in Fig 5?

The data sources need a greater explanation in the methods section. In particular, it is unclear what 10-year bond yields were used for which countries as two sources are provided in Supplementary 2 as well as “web search” (what does this mean?). Similarly, two sources are provided for fossil fuel sales.

Discussion

“Many developed countries... Permanent CDR measures serve as a promising way to pull down fair emission allocation overshoot.” I feel this part of the discussion understates the implications. Really what the findings suggest is that in the interests of equity, developed countries should be aiming to go beyond net zero and CDR should be seen more as a tool for going beyond rather than meeting net zero.

For greater balance considering it is a fairly polemic topic, I would like to see the description of CDR as a “promising way” changed to something more neutral like “potential” and a brief mention of the potential risk CDR poses if developing countries are over-reliant on it as a strategy at the expense of cutting current emissions.

This paper could also be touched on <https://doi.org/10.1016/j.jenvman.2022.116211> as CDR potentials vary among countries and some countries may not be able to compensate for their fair emission overshoot through CDR alone. This might strengthen the need for financial support, technology transfer etc., touched on in the final sentence of the discussion – the discussion could also touch on the relevance of this for the greater support needed by developed countries to help developing countries achieve their conditional NDCs.

The analysis uses GWP* for methane emissions, which presents the implication that countries may be able to compensate for some of their fair emission overshoot through methane reductions in the future. This should also be touched on in the discussion as an alternative compensation strategy to CDR and financial support, particularly as a lot of the novelty in this paper is around the inclusion of non-CO2 gases.

Minor points

The heading “results” comes too early in the paper.

When discussing the numbers of countries with negative/positive remaining budgets, it is not clear why ranges are presented (what uncertainty is this capturing).

The caption of Figure S9 mentions 1.5°C when it should be 2°C.

The authors mention uncertainties in historical contributions being a limitation of the study. Particularly relevant here are potential inconsistencies with LULUCF emissions in the PRIMAP data, which should be highlighted. I would also recommend explicitly mentioning in the methods section when the PRIMAP data is discussed that LULUCF is included instead of just saying that it “contains all sectors” as LULUCF is not included in the main version of the dataset.

The manuscript could be clearer that it is based on production-based emissions and expand more on the implications if consumption-based emissions were used instead. An alternative to the use of the word “consumption” when referring to carbon budgets could also be considered to avoid possible misinterpretation.

The scale of the y-axis in Supplementary Fig. 4 could be changed to improve interpretability as the figure is largely white space in its current form.

(Remarks to the Author)

This article is a well-justified and elaborated addition to the literature on equity in the context of the Paris Agreement. It extends prior work by considering the role of non-CO₂ greenhouse gas emissions and their contributions to warming. The results are not unexpected: contributions to warming are concentrated among developed nations, in contrast to least developed and some developing nations. If we push the analysis forward to 2050, it is likely that developed nations will significantly exceed their allocations. But despite this lack of novelty, this article does set a new standard in the existing (and large) literature on international climate equity by linking its methodology to warming contributions, and in my opinion should be published with revisions.

My first main comment concerns the distinction between the 'equity indicators' and the 'equity principles'. On my first reading, it was not clear to me what the exact difference was. The section starting on line 69 makes it clear that three 'equity principles' are developed, which are combinations of specific indicators shown in figure 1 (note: I suggest giving these three principles names for easier referencing). However, in subsequent sections (figure 3 onwards), the focus turns back to the underlying indicators, not the 'equity principles'. Confusingly, coloured pies are shown next to each indicator in the figures, meant to represent the 'equity principles', even though each interpretation is a combination of indicators. This problem is repeated in figure 5 and figure 6, at which point I do not know whether to focus on the indicators, or their combinations in the 'equity principles'. Would it not be simpler to focus on the principles only, using a composite indicator for each?

A second main comment concerns the notion of a "deepest technologically feasible" pathway. Given that this is a paper on justice and equity, it is surprising to see this arrive in the manuscript in its current form. First of all, it is not fully clear in the main text where these pathways come from. My reading of the supplementary materials is that these are the country or region specific IAM results averaged across the AR6 C1 scenario category. I suggest explicitly stating this in the methods section of the main manuscript. Second, by the authors own definition, are these the "deepest technologically feasible" pathways? Would that not be the country or region specific pathway in the AR6 database with the lowest cumulative emissions? (or, e.g. the lowest quartile of cumulative emissions?). Third, I would revisit the definition itself. What does 'feasible' mean, and how did the authors judge this? Do they also consider the feasibility of, for example, CDR scaling, and rule out those that exceed biomass availability constraints? Fourth, it is not acknowledged that these are techno-economic pathways that already carry the weight of an implicit but highly disputed equity principle: "least-cost". Simply comparing the shapes of the grey curves shows that Brazil and China have extraordinarily steep pathways, but the EU and USA do not. Does it mean that it is technologically infeasible for the EU and USA to reach net negative CO₂ emissions before 2050? Or simply that it is cheaper to do so in other regions? I would strongly recommend to search for a different wording than "deepest technologically feasible", unless the authors plan to open up all these different points.

More specific comments as follows:

Page 2, line 48: another way to justify this further study on allocation procedures could be to set it in the context of climate damages and assistance, as for example in Fanning and Hickel 2023 (<https://www.nature.com/articles/s41893-023-01130-8>).

Page 2, line 56: it is not clear why GWP* is specifically needed here to convert GHGs into CO₂ equivalents. Actually, the rest of the manuscript does not explain this choice at all, which is surprising given that other metrics could also be selected. I suggest elaborating on this at some.

Page 2, line 72: figure 1 does not depict the stated elements in this sentence ("do no harm" and "precautionary principles")

Page 3, Figure 1: if one quickly scans this figure without reading the text below it, the interpretation is that all equity principles and indicators are applied to resolve a final allocation principle obtained in the bottom right box. I suggest re-drawing to avoid this interpretation. If the analysis focuses on the first/second/third 'equity principles', these should be upfront in the visualisation.

Page 5, line 155: how do countries follow fair allocations from 2021 if their allocation is already exhausted (as stated in the prior paragraph)?

Page 6, Figure 3: based on the prior comment, extending this figure axis to 2050 would show how the post-2021 'projection' is arranged for each country. Beyond this, I find the mixture of indicators and equity principles confusing. The colours used to introduce each indicator in figure 1 are not carried on here, leading to a mismatch between the lines and the pies. Also the pies refer to multiple indicators, but are assigned to individual ones here. Also note that "1900" instead of "1990" is annotated in the top left panel.

Page 7, Figure 4: I am not sure how useful this figure is. From it, I can learn that many LDCs and developing countries have small per capita CO₂-we emissions. Is this important to know? In the right panel, I confess to having no idea. The per capita per year budget stays stable from 2022-2049: one might interpret this to mean that emissions are 0 during these years, which is probably not the intention of the figure?

Page 8, Figure 5: same problem as figure 3 that there is a mismatch of the line and pie colours with figure 1.

Methods: is PRIMAP Hist-CR or Hist-TP used?

Supplementary figure 5: this is a very insightful figure that directly relates to discussion text in the main manuscript. I would

suggest to use it instead of figure 4.

Version 1:

Reviewer comments:

Reviewer #2

(Remarks to the Author)

I thank the authors for their thorough response to my previous review. I have one outstanding concern and a few minor suggestions:

1. In the Methods, the authors now clarify the emission data they use cover CO₂, CH₄, N₂O, and F-gases, excluding AFOLU. Presumably, this is a mistake and should be LULUCF instead of AFOLU as the manuscript implies agricultural methane emissions are included "The approach in this study ensures that historical warming contributions of countries with large non-CO₂ emissions, such as Brazil and New Zealand, are accurately accounted for."

Beforehand there is also the implication that this study includes LULUCF emissions (which is what I had assumed on my first reading) when talking about an advantage of this study over others "Typically, only CO₂ emissions excluding LULUCF are allocated". I am still unsure whether the CO₂-we analysis includes LULUCF. This becomes more unclear after seeing Figure 5a, which compares CO₂-we budgets to CO₂ excluding LULUCF and CO₂ including LULUCF. LULUCF emissions were therefore included for at least part of the analysis even if it just the comparison in Figure 5a.

- The main text needs to be clear about whether LULUCF emissions were included or not in the CO₂-we analysis. If they are not included, it would be good to explain why not and justify their inclusion in Fig 5a. Lines 325-331 should also be more carefully written to not imply that a strength of this study over others is that they typically do not include LULUCF.

- LULUCF data (other than the exclusion of AFOLU) is not mentioned in the methods nor are the other choices in Figure 5 (e.g. are all sectors still included in the CO₂ comparisons) The use of LULUCF and the treatment of emissions in Fig 5a needs a clear explanation in the Methods along with a mention of the limitations of the PRIMAP LULUCF data.

- As a more minor point, it would make sense to reorder Fig 5a so CO₂-we is on the left as this is the base case in the rest of the analysis.

The exclusion or inclusion of LULUCF has a considerable effect on the medium and high-development countries' budgets in Fig5a (greater than whether non-CO₂ gases are included by an eye test) so this point needs more attention and clarification.

Minor suggestions and clarifications:

2. Lines 34-38: The descriptions of the Paris pledges could be explained more clearly. Using "many governments" does not imply the "near-global coverage" of pledges. It would also be good to distinguish between NDCs and LTSs as there is not a near-global coverage of LTSs.

3. I still find the heading "Results" before "Interpretation of equity principles" strange as this reads more like a methods section. It would make more sense to me to have "Results" before the "Fair national CO₂ warming equivalents" section. This may be a point for the Editor's input as the previous response referred to the Nature Communications guidelines.

4. I find the new inclusion of different equity interpretation names (e.g. Interpretation I) helpful. It would be nice to revise the "calculation of the allocation procedure" section of the Methods to also include these names for consistency and perhaps add subheadings to help guide readers when reading this section.

5. Line 93: The justification for 1990 being used (i.e. excusable ignorance) is mentioned later in the manuscript but not here where 1990 is first mentioned. I would move the justification here to help explain this choice on its first mention.

5. Fig 1: What does "before peak" mean? Is this before peak warming? There is potential for misinterpretation (e.g. before peak emissions) given the context.

6. Lines 127-128: better to add something along the lines of "as defined by UNDP HDI ranking".

7. Lines 166-174: In response to point 13 of my previous review, the authors state that they no longer report ranges from the remaining budgets. However, ranges are still reported in lines 166-174 in the revised manuscript. These should either be changed to single values or the uncertainty in the ranges clarified.

8. Line 275-276: South Korea is not really a good example of a country relying "primarily on the tertiary sector" as they have a strong industrial sector. It is also a bit of an exaggerated statement for Japan and Switzerland as about a quarter of their GDPs are from industry.

9. Finally, the different equity interpretations have a considerable focus in the paper but do not have much influence on the

overall rankings of countries (the coverage of gases and sectors seems much more significant). Perhaps the authors could comment a bit more on their interpretation of this. Do they feel that the differences arising from different equity interpretations are small enough that the most simple Interpretation I is sufficient?

Reviewer #3

(Remarks to the Author)

The authors have adequately addressed my comments in the first round of review and I would be happy to now see this piece published.

Version 2:

Reviewer comments:

Reviewer #2

(Remarks to the Author)

The authors have addressed the remaining issues from my last review so I am now happy to recommend publication.

We appreciate the reviewers' valuable comments and constructive suggestions. We have carefully revised the manuscript according to these comments. Point-to-point responses are provided below. The reviewers' comments are **in black**, our responses are **in blue**, and the corresponding revises in manuscript are **in orange**. We also include page and line numbers in the documents for editor and reviewers.

=====

Reviewer #1:

=====

I find this to be a high-quality study touching on an important topic. The level of novelty for publication in Nature Communications is debatable as it is not exactly the first paper to look at fair shares of decarbonisation pathways. While the inclusion of non-CO2 and LULUCF emissions does make it a more complete study than others, these additions do not really contribute an important advance that substantially changes the overall picture.

Nonetheless, I would overall support its publication (subject to revisions outlined below) as I find the study to be very effective at drawing attention to the equity considerations of global climate policy and would have broad interest.

We deeply appreciate the positive comment from the reviewer.

General comments

1. The manuscript focuses on a 1.5°C budget with the 2.0°C budget relegated to the supplementary information with only a brief mention in the main text that a similar picture emerges for 2°C. I feel like the paper would be more impactful if this was flipped and it primarily focussed on 2.0°C. Firstly, many consider the 1.5°C target to be now be unrealistic so some readers might dismiss this analysis as only applying to an unrealistic scenario and overlook the important finding that the implications are essentially the same for 2°C. Secondly, the current state of NDCs and LTSs suggests around 2°C so focusing on 2°C would make the study more relevant to the current state of global climate pledges. This change would also emphasise the point that even though the Paris Agreement has been successful at producing pledges in line with the overall 2°C target target there remain equity issues even if countries achieve their pledges (this aspect could also be elaborated on more in the Introduction).

For these reasons, I would find the article more interesting if it focused on 2°C but this is more of a personal opinion on what would make more impactful research rather than a flaw in the paper. In either case of the authors changing the focus to 2°C or staying with 1.5°C, the other budget needs far greater discussion in the main text other than just "a similar picture emerges for 2°C". It is not immediately obvious and intuitive the picture would be so similar considering that the remaining 2°C budget is roughly 5 times larger than the remaining 1.5°C budget.

Thank you for raising the discussion regarding the focus of this research on the global warming budget. We understand and respect your suggestion to focus the main text on 2°C rather than the 1.5°C. However, we have chosen to center our analysis on the 1.5°C target for several reasons: Firstly, while the 1.5°C target may appear increasingly challenging to meet, it remains focal point of the negotiations. Paris Agreement is about limiting warming "well below 2°C" while pursuing to

limit it to 1.5°C, the COP26 Glasgow Climate Pact and COP27 Sharm El-Sheikh outcomes further put emphasis on 1.5°C. Secondly, by focusing on 1.5°C target, we aim to highlight the urgent need for more ambitious climate actions. Thirdly, we have added analysis to provide a robust comparison under 2°C, demonstrating that the implications for 2°C are similar to 1.5°C. When adjusting the warming target from 1.5°C to 2°C, the qualitative implications of this study, as outlined in the first paragraph of the discussion, remain unchanged. Although the quantitative results will experience some changes, countries' budgets show only minor changes in rankings. We believe this approach will maintain the integrity of our primary focus while addressing broader concerns.

That said, we would like to keep 1.5°C as the global warming budget in the main text and include further comparison with the outcomes under 1.5°C, 1.7°C, and 2°C in the session “Influence of uncertainty due to methodological factors and aerosol masking effect” (see Page 12, Line 304-314) and “Supplementary Note 6: Influence of different desired peak global warming target” (see SI Page 9-12). We have revised the manuscript to provide a more detailed discussion of the 2°C budget, ensuring that readers clearly understand the implications for both temperature targets.

“For instance, adjusting the desired peak global warming target from 1.5°C to 2°C increases the remaining warming budgets for all countries. Countries whose remaining budgets shift from below zero under 1.5°C to above zero under 2°C, include South Africa, Israel, Iran, Cuba, North Korea, China, and Botswana. However, this adjustment's influence is less significant than the variability introduced by different estimates of the aerosol masking effect, comparable to the influence of adjustment extent, and barely noticeable compared to the substantial negative budget for major countries with very high human development. Additionally, countries' budgets show only minor changes in rankings. No country's budget shifts from below the global average to above it, and only Chile and Niue see their budgets move from above to below the global average. This suggests that qualitative implications of this study remain unchanged, with a clear distinction in remaining budgets among country groups (Supplementary Note 6, Supplementary Fig. 8-11).” (see Page 12, Line 304-314)

2. It would also be useful to have some kind of summary table or figure to better illustrate the differences between the various scenarios and sensitivity tests. Figure 2 of the main text and Supplementary Figure 9 at first appear identical until spotting the big differences in the global average remaining budgets per capita. As a possible example, a table providing the average remaining budgets by country groupings might help with this. A similar issue arises with Supplementary Figure 7 as it is impossible to notice the difference this change in assumption has made to the results compared to Figure 2

Thank you for your suggestions! Supplementary Fig. 9 differs in the global remaining CO₂-we budgets compared with Fig. 2. The national total (shown as the height of bars above the x-axis) and remaining CO₂-we budgets (shown as the step lines) differs accordingly, yet the country rankings are almost the same. Supplementary Fig. 7 differs in the adjustment extent of national total CO₂-we budgets by the driving indicator compared with Fig. 2. Thus, the national remaining CO₂-we budgets under Interpretation II and Interpretation III (shown as the grey step lines) differ, the country

rankings are almost the same.

We added a Supplementary data table (see Supplementary data 4_average remaining CO₂-we budgets by country groupings across sensitivity test) providing the average remaining budgets by country groupings across sensitivity tests. A short example is shown below.

Table 1 Average remaining CO₂-we budgets by country groupings across sensitivity tests

Equity interpretation	Warming target (°C)	Adjustment extent	Start year of historical contributions	Aerosol masking effect percentile	Average remaining CO ₂ -we budgets per capita (2022-2050) (tCO ₂ -we)				
					Low human development	Medium human development	High human development	Very high human development	Not classified
Interpretation II-A	1.5	50%	1850	0.5	453	311	19	-957	-43
Interpretation III	1.5	50%	1850	0.5	352	316	59	-908	-45
Interpretation II-B	1.5	50%	1850	0.5	397	348	53	-1008	-59

3. The manuscript also does not sufficiently describe the categorisation of “developed” and “developing” countries and LDCs. The use of the M49 classification is not ideal as the distinction is not clearly defined (see “note on developed and developing regions” on the M49 webpage including in the authors’ references. A better approach would be to classify the countries either by UNDP HDI tiers (very high, high, medium, and low) or World Bank country classifications by income (High, upper-middle, lower-middle, and low).

In my opinion, it would also benefit the results to have four groups instead of three (based on the HDI or income categories) as currently the “developing” countries group is very large and puts countries at very different stages of development together.

Thank you for your suggestions of alternative categorizations. We acknowledge the dispute around categorization of developed countries, developing countries and LDCs. We’ve changed the classification to UNDP HDI tiers in the illustration in the main text. We have edited the corresponding figures (see Page 5, Fig.2 and Page 9, Fig.5) and the accompanying analysis in the main text (see Page 4, Line 127-146).

Under the classification of UNDP HDI tiers, the national remaining CO₂-we budgets show a relatively clear distinction between very high, high, and medium/low human development countries. Yet, the differentiation between medium and low human development countries is less pronounced. Please see Page 4, Line 127-146 for further details.

Fig. 2 Total (1850-2050), historical (1850-2021), and remaining (2022-2050) CO₂ warming equivalent budgets per capita by country. Countries are ranked according to their remaining per capita national budgets, under the central case which considers a historical warming contribution starting in 1850 and a global warming target of 1.5°C. The height of bars above the x-axis represents the total CO₂-we budgets per capita under equity Interpretation I that considers the equality and polluter-pays principles. The height of the bars below the x-axis represents the historical consumed CO₂-we budgets per capita. The width of the bars represents a country's population in 2050. The step lines represent the remaining CO₂-we budgets per capita per country with the black line referring to Interpretation I and the grey lines referring to Interpretation II-A, II-B, and III. Very high, high, medium, and low human development countries are marked in different colours.

We leave the World Bank income classification, along with the original categories of developed countries, developing countries and LDCs, in the Supplementary Information for the readers' reference and interest. We have added some analysis in "Supplementary Note 8: Illustration of alternative country categorization method" (see SI Page 14-16, Supplementary Fig.14 and Supplementary Fig.15) to show the difference between alternative country categorization methods.

Supplementary Fig. 12 Total (1850-2050), historical (1850-2021), and remaining (2022-2050) CO₂ warming equivalent budgets per capita by country. Countries are ranked according to their remaining per capita national budgets, under the central case which considers a historical warming contribution starting in 1850 and a global warming target of 1.5°C. The height of bars above the x-axis represents the total CO₂-we budgets per capita under equity Interpretation I that considers the equality and polluter-pays principles. The height of the bars below the x-axis represents the historical consumed CO₂-we budgets per capita. The width of the bars represents a country's population in 2050. The step lines represent the remaining CO₂-we budgets per capita per country with the black line referring to Interpretation I and the grey lines referring to Interpretation II-A, II-B, and III. High, upper middle, lower middle and low income countries are marked in different colours.

Supplementary Fig. 13 Total (1850-2050), historical (1850-2021), and remaining (2022-2050) CO₂ warming equivalent budgets per capita by country. Countries are ranked according to their remaining per capita national budgets, under the central case which considers a historical warming contribution starting in 1850 and a global warming target of 1.5°C. The height of bars above the x-axis represents the total CO₂-we budgets per capita under equity Interpretation I that considers the equality and polluter-pays principles. The height of the bars below the x-axis represents the historical consumed CO₂-we budgets per capita. The width of the bars represents a country's population in 2050. The step lines represent the remaining CO₂-we budgets per capita per country with the black line referring to Interpretation I and the grey lines referring to Interpretation II-A, II-B, and III. Developed countries, developing countries and least developed countries (LDCs) are marked in different colours.

4. The manuscript also does not state the overall size of the calculated remaining budgets. This should be mentioned somewhere so that the readers can see that it is consistent with other studies e.g. (Lamboll et al. 2023).

Thanks for your kind request for clarification! We use warming contribution as the allocation object. The warming contribution is then converted into CO₂-we budgets using the Transient Climate Response to Cumulative Emissions (TCRE), which could be compared with remaining carbon budgets.

In our calculation, the historical warming contribution equals 1.22°C by 2021. The size of calculated global remaining budgets from 2022 is 282 Gt under 1.5°C and 1232 Gt under 2°C, which is consistent with the estimation of Lamboll et al¹ (remaining carbon budget for a 50% chance of

keeping warming to 1.5 °C is around 250 GtCO₂ as of January 2023. For a 50% chance of 2 °C the remaining carbon budget is around 1,200 GtCO₂).

We've added and explained the overall size in the method "Historical warming contribution": "Under this calculation, the historical warming contribution equals 1.22°C by 2021. The size of the calculated global CO₂ warming-equivalent emission budget from 2022 equals 282 GtCO₂-we for a median global warming limit of 1.5°C and 1232 GtCO₂-we for 2°C, consistent with the estimation of previous studies¹." (see Page 16, Line 458-462)

Methods

5. The population assumptions are not clear – presumably, the authors used the UN's medium scenario but this is not stated.

The projected population to 2050 is a key assumption for allocating the budgets so sensitivity analysis using the UN's high- and low-fertility scenarios should be provided.

Thanks for your kind request for clarification! For population data, we followed United Nations medium scenario projection, a scenario with medium fertility, medium mortality and medium international migration. We've clarified the scenario we used for population assumptions in the Method section (see Page 17, Line 522-523 and Supplementary Table 3).

We acknowledge that population projections can influence the numerical outcomes. However, we believe that there is little that can be done from a policy perspective to influence these pathways over this short timeframe. Therefore, we have adhered to the medium-fertility scenario in the main text. Nonetheless, we have conducted a sensitivity analysis using the UN's high- and low-fertility scenarios, which is now detailed in "Supplementary Note 7: Sensitivity to population projection" (see SI Page 12-14), Supplementary Fig. 12 and Supplementary Fig. 13.

"For instance, under high-fertility population projections, countries experience a reduction in their total CO₂-equivalent warming budgets per capita due to increased population compared with medium-fertility population projections. This effect is more pronounced in countries with medium and low human development, as these countries' populations are more affected. However, when considering the remaining CO₂-equivalent warming budgets per capita, the impact of population projection appears to be marginal." (see SI Page 12, Line 181-186)

Supplementary Fig. 12 Influence of population projection on CO₂ warming equivalent budgets. Total (1850-2050) (a) and remaining (2022-2050) (b) CO₂ warming equivalent budgets per capita, compared when using population projection under both medium and high fertility scenarios. All allocations adhere to the equity Interpretation I. The size of bubbles indicates the amount of national emissions in 2021. Very high, high, medium, and low human development countries are marked in different colours.

Supplementary Fig. 13 The uncertainty in remaining (2022-2050) CO₂ warming equivalent budgets for major countries, with various population projections. All allocations adhere to the equity Interpretation I. The x-axis shows different population projections due to fertility. Connected points represent data from the same country, showing how a country's budget changes as population projection is changed. The overlapped density plots depict the distribution of remaining budgets by country group.

6. It is not clear which countries have been included in the analysis – is this all parties to the UNFCCC? I do not think the overall N is stated in the manuscript and there is no explanation of which countries may have been included or excluded.

The list of countries in the analysis add up to 195, covering all UNFCCC member states with the exception of the Holy See and the State of Palestine.

We clarified the list of countries in the Method, see Page 13, Line 381-382, and the overall size of N, see Page 14, Line 401.

We added a supplementary data table for a list of all countries in the analysis, and their corresponding category (see Supplementary data 5_ national raw data for indicators).

7. The EU27+UK countries appear together in Fig. 5 but seem to have been treated individually elsewhere as some countries are referred to individually. This should also be made clearer – is it a case of combining them just for illustrative purposes in Fig 5?

The EU27+UK countries are treated individually through the calculation process, and are combined only for illustrative purposes. We add a note to clarify this in the caption of former Fig.5.

“EU27+UK countries are treated as individual entities through the fair allocation process, and are combined only for illustrative purposes.” (see Page 12, Line 212-214)

8. The data sources need a greater explanation in the methods section. In particular, it is unclear what 10-year bond yields were used for which countries as two sources are provided in Supplementary 2 as well as “web search” (what does this mean?). Similarly, two sources are provided for fossil fuel sales.

We have added “Supplementary Note 9: Description of data sources and data washing process” (see SI Page 17-19) to provide a detailed explanation of the data sources and our methodology for integrating data from multiple sources to create a unified base dataset for all countries. We've revised Supplementary Table 1 Data sources used in this study (see SI Page 17, Line 270) to explain the data details.

To summarize, For 10-year bond yield, data are sourced from two datasets^{2,3} and individual web searches. For countries with data available from both datasets, we adopt the average value. For countries with no data in both datasets, we conduct web searches for the countries with the top 30 populations or top 30 emissions worldwide. We've added Supplementary Table 2 Country included in each data source for 10-year bond yield (see SI Page 17, Line 282) to list the precise web link

where we retrieved 10-year bond yield data in each source.

For fossil fuel sales, national historical cumulative fossil fuel sales are calculated by summing the products of the prices and production volumes of coal, oil, and gas, aggregated cumulatively from 1900 to 2016. Multiples sources are used in this process, and each one is elaborated. “For fossil fuel sales, national historical cumulative fossil fuel sales are calculated by summing the products of the prices and production volumes of coal, oil, and gas, aggregated cumulatively from 1900 to 2016. Coal, oil, and gas production data from 1900 to 2016 are derived from Ourworldindata⁴ which combines information from the Energy Institute Review of World Energy⁵ and ref.⁶. Coal, natural gas price from 1850 to 2020 are derived from D. S. Jacks⁷. Historical crude oil prices from 1861 to 2023 are derived from ChartsBin⁸ showing data from the BP Statistical Review of World Energy^{9, 10}.” (see SI Page 18, Line 284-290)

Discussion

9. “Many developed countries... Permanent CDR measures serve as a promising way to pull down fair emission allocation overshoot.” I feel this part of the discussion understates the implications. Really what the findings suggest is that in the interests of equity, developed countries should be aiming to go beyond net zero and CDR should be seen more as a tool for going beyond rather than meeting net zero.

For greater balance considering it is a fairly polemic topic, I would like to see the description of CDR as a “promising way” changed to something more neutral like “potential” and a brief mention of the potential risk CDR poses if developing countries are over-reliant on it as a strategy at the expense of cutting current emissions.

We agree with you that we should be careful of the language regarding CDR.

We changed the wording from “promising way” to “potential way”, to emphasize that CDR is the last resort to achieve fair allocation quotas, after countries have done their best to reduce to the maximum amount.

“Permanent CDR measures serve as a potential way to pull down fair emission quota overshoot, as well as to address the uncertain climate response¹¹.” (see Page 13, Line 365-367)

We further extended the consideration of several trade-offs of CDR, potential risk of over-relying on CDR as well as the need to rapidly build preventative CDR capacity to deal with uncertain climate response.

“Yet, CDR often comes with side effects that impact economic prosperity, well-being, environmental, and human health, with negative consequences possibly outweighing the positive¹². Therefore, it is crucial that CDR is not overly relied upon at the cost of substituting short emission reductions or threatening other development goals¹³.” (see Page 13, Line 368-371)

10. This paper could also be touched on <https://doi.org/10.1016/j.jenvman.2022.116211> as CDR potentials vary among countries and some countries may not be able to compensate for their fair emission overshoot through CDR alone. This might strengthen the need for financial

support, technology transfer etc., touched on in the final sentence of the discussion – the discussion could also touch on the relevance of this for the greater support needed by developed countries to help developing countries achieve their conditional NDCs.

We added additional rationale for financial support, technology transfer, or capacity building: “Considering the unevenly distributed country-level CDR deployment potentials, overemitting countries may find themselves unable to compensate for their fair emission quota overshoot solely through domestic CDR¹⁴.” (see Page 13, Line 371-373)

We addressed the benefit of international support on conditional NDCs in the discussions. “Such international support, particularly climate finance, could also help developing countries to implement their conditional NDCs, further improving equity and enhancing ambition¹⁵.” (see Page 13, Line 375-377)

11. The analysis uses GWP* for methane emissions, which presents the implication that countries may be able to compensate for some of their fair emission overshoot through methane reductions in the future. This should also be touched on in the discussion as an alternative compensation strategy to CDR and financial support, particularly as a lot of the novelty in this paper is around the inclusion of non-CO₂ gases.

This research allocates the national fair share based on the object of the global allowable warming effect. Both CO₂ and non-CO₂ GHGs are included in the fair share, and countries could determine their pathway to meet the fair share, including the type of GHGs. Countries should always first consider reducing to the maximum effort, including methane reduction, before using alternative ways to compensate for their fair emission overshoot.

Methane is a topic attracting increasing focus, as it is the second largest contributor to radiative forcing historically. By using GWP*, we include non-CO₂ GHGs in countries’ warming contributions, making non-CO₂ (like methane) reduction one of the choices to achieve fair shares.

We address the importance of methane reductions as,

“Substantial global non-CO₂ GHG reductions (for instance CH₄) offers the potential to limit peak warming and prevent surpassing critical climate thresholds. If appropriately accounted for such reductions also help to limit the rapid depletion of fair shares^{16, 17}.” (see Page 12, Line 331-334)

Minor points

12. The heading “results” comes too early in the paper.

Thank you for your feedback regarding the positioning of the “Results” heading. We understand that you feel it may come too early in the paper. We would like to clarify and provide some context for our decision:

Firstly, according to the format of Nature Communications, the text must be divided into the specific sections: “Introduction, Results, Discussion (optional), Methods (optional)”. It is therefore necessary to include a separate Results section in the main text, and Results could be further divided into

subsections.

Secondly, the establishment of the allocation framework is a foundational component of our study achievements, and we believe it is appropriate to introduce it at the beginning of the Results section. This approach aligns with previous research^{18, 19} on fair sharing topics, where frameworks were presented initially in the Results section.

Given these considerations, we prefer to retain the “Results” heading in its current position. However, we are open to further suggestions if you have specific concerns about the structure.

13. When discussing the numbers of countries with negative/positive remaining budgets, it is not clear why ranges are presented (what uncertainty is this capturing).

Thank you for providing the feedback for the confusion caused by the ranges.

The range was intended to indicate the uncertainties across all three equity interpretations. Equity interpretations reflect varying understandings of the equity concept during detailed quantification processes.

To avoid potentially ambiguous implications, we have opted not to present the results across different equity interpretations as ranges in the main text. Instead, we have selected equity Interpretation I as the central case to present in the main text. For instance, we now express the results as single values: “Of 68 very high human development countries, 59 are left with negative remaining budgets for the 2022-2050 period under equity Interpretation I, with historical warming contributions starting in 1850 and a global warming target of 1.5°C” (see Page 4, Line 129-132). The variation in quantification results when applying various equity interpretations are illustrated in Fig. 2, where the grey step lines correspond to Interpretation II-A, II-B, and III. They are also illustrated by the coloured lines in Fig. 3 and Fig.5, and along the x-axis in Fig. 5b.

14. The caption of Figure S9 mentions 1.5°C when it should be 2°C.

Thank you for pointing out the error. The caption of previous Supplementary Fig. 9 has been corrected to indicate 2°C (see SI Page 12, Supplementary Fig. 11).

15. The authors mention uncertainties in historical contributions being a limitation of the study. Particularly relevant here are potential inconsistencies with LULUCF emissions in the PRIMAP data, which should be highlighted. I would also recommend explicitly mentioning in the methods section when the PRIMAP data is discussed that LULUCF is included instead of just saying that it “contains all sectors” as LULUCF is not included in the main version of the dataset.

Thank you for this nice reminding. When calculating historical contributions, we use the national total which contains sectors other than Agriculture, Forestry and Other Land Use (AFOLU) and international aviation and shipping.

We’ve clarified it in the method, “In this study, the GHG gases include CO₂, CH₄, N₂O, and F-gases, and we use the national total that contains all sectors other than Agriculture, Forestry and Other

Land Use and international aviation and shipping.” (see Page 15, Line 441-443)

16. The manuscript could be clearer that it is based on production-based emissions and expand more on the implications if consumption-based emissions were used instead. An alternative to the use of the word “consumption” when referring to carbon budgets could also be considered to avoid possible misinterpretation.

Thank you for this nice reminding. This study is conducted based on production-based emissions. In the discussion section, we’ve expanded the limitation regarding consumption-based emissions: “Second, the quantification of countries’ historical contributions is based on production-based emissions, that is, emissions within a national territory. Emission transfers through international trade add to the overall uncertainty. Considering value-chain or consumption-based emissions tends to increase developed countries’ historical warming contributions²⁰, although also low-income countries without established industrial sectors are affected.” (see Page 13, Line 346-350) The word “consumption” when referring to carbon budgets has been deleted or replaced. For instance, we’ve revised the subtitle “Consumption of remaining budgets” to “Depletion of remaining budgets” (see Page 5, Line 163).

17. The scale of the y-axis in Supplementary Fig. 4 could be changed to improve interpretability as the figure is largely white space in its current form.

The y-axis scale in previous Supplementary Fig. 4 has been adjusted to reduce the white space (see SI Page 6, Supplementary Fig.5).

Supplementary Fig. 5 Total (1850-2050), historical (1850-2021), and remaining (2022-2050) CO₂

warming equivalent budgets per capita by country (starting year of historical warming contribution of 1990). Countries are ranked according to their remaining per capita national budgets, under the central case which considers a historical warming contribution starting in 1850 and a global warming target of 1.5°C. The height of bars above the x-axis represents the total CO₂-we budgets per capita under equity Interpretation I that considers the equality and polluter-pays principles. The height of the bars below the x-axis represents the historical consumed CO₂-we budgets per capita. The width of the bars represents a country's population in 2050. The step lines represent the remaining CO₂-we budgets per capita per country with the black line referring to Interpretation I and the grey lines referring to Interpretation II-A, II-B, and III. Very high, high, medium, and low human development countries are marked in different colours.

=====
Reviewer #2:
=====

This article is a well-justified and elaborated addition to the literature on equity in the context of the Paris Agreement. It extends prior work by considering the role of non-CO₂ greenhouse gas emissions and their contributions to warming. The results are not unexpected: contributions to warming are concentrated among developed nations, in contrast to least developed and some developing nations. If we push the analysis forward to 2050, it is likely that developed nations will significantly exceed their allocations. But despite this lack of novelty, this article does set a new standard in the existing (and large) literature on international climate equity by linking its methodology to warming contributions, and in my opinion should be published with revisions.

1. My first main comment concerns the distinction between the 'equity indicators' and the 'equity principles'. On my first reading, it was not clear to me what the exact difference was. The section starting on line 69 makes it clear that three 'equity principles' are developed, which are combinations of specific indicators shown in figure 1 (note: I suggest giving these three principles names for easier referencing). However, in subsequent sections (figure 3 onwards), the focus turns back to the underlying indicators, not the 'equity principles'. Confusingly, coloured pies are shown next to each indicator in the figures, meant to represent the 'equity principles', even though each interpretation is a combination of indicators. This problem is repeated in figure 5 and figure 6, at which point I do not know whether to focus on the indicators, or their combinations in the 'equity principles'. Would it not be simpler to focus on the principles only, using a composite indicator for each?

Thanks for raising this important question. The fair allocation framework in this study is built upon three layers. Firstly, four equity principles are derived from a general ethical background, which are principles for equitable effort sharing discussed in the climate equity literature. Secondly, each equity principle is translated into one or two indicators for further quantification in the allocation procedure. Thirdly, three equity interpretations are formed from combinations of equity principles. Specifically, Interpretation II incorporates the "Ability to Pay" principle, which is quantified using two

indicators. Consequently, Interpretation II is further subdivided to Interpretation II-A and Interpretation II-B based on the different indicators included.

Throughout the subsequent results, the presentation of figures and texts is now focused on equity interpretations, with Interpretation I as the central case to present in the main text. We have made below adjustment to ensure readers' focus are now on equity interpretations:

We have revised Fig. 1 (see Page 4, Fig.1) to explicitly include these three layers: equity principles, indicators, and equity interpretations. The progression from left to right illustrates the formation process through which the equity interpretations are ultimately constructed. Each interpretation is represented by a distinct color, and given names for easier referencing.

- ① Interpretation I relies on the principles of equality and polluter pays.
- ② Interpretation II-A and Interpretation II-B introduce the principle of ability to pay to equality and polluter pays.
- ③ Interpretation III introduces the principle of beneficiary pays to equality and polluter pays.

Fig. 1 Analysis framework linking equity principles, indicators, and the allocation procedure to estimate fair global warming contributions. Each equity principle and the corresponding indicators are marked in different colours: pink stands for the principle of equality, light blue for ability to pay, light green for beneficiary pays and yellow for polluter pays. Interpretations are marked in different colours: dark blue stands for Interpretation I, orange for Interpretation II-A, dark green for Interpretation II-B and red for Interpretation III. The horizontal coloured arrows illustrate the incorporation of equity principles and indicators into the allocation procedure. Under the “Indicators” column, plus (or minus) signs in brackets indicate the indicators are directly (or inversely) proportional to the remaining national allowable warming budgets. Under the “Interpretation” columns, the vertical grey arrows suggest whether the corresponding indicator is included in this step.

The coloured lines in Fig.3 and Fig.4 (the previous Fig.5) represent equity interpretations, with color and labels consistent to Fig.1. The legend in Fig.3 and Fig.4 have been changed accordingly. We have removed the pies in Fig.3 and Fig.4 (the previous Fig.5) to avoid confusion.

We highlighted that the primary focus of main text is on equity Interpretation I when presenting the first result: “Of 68 very high human development countries, 59 are left with negative remaining budgets for the 2022-2050 period under equity Interpretation I, with historical warming contributions starting in 1850 and a global warming target of 1.5°C” (see Page 4, Line 129-132).

2. A second main comment concerns the notion of a “deepest technologically feasible” pathway. Given that this is a paper on justice and equity, it is surprising to see this arrive in the manuscript in its current form. First of all, it is not fully clear in the main text where these pathways come from. My reading of the supplementary materials is that these are the country or region specific IAM results averaged across the AR6 C1 scenario category. I suggest explicitly stating this in the methods section of the main manuscript. Second, by the authors own definition, are these the “deepest technologically feasible” pathways? Would that not be the country or region specific pathway in the AR6 database with the lowest cumulative emissions? (or, e.g. the lowest quartile of cumulative emissions?). Third, I would revisit the definition itself. What does ‘feasible’ mean, and how did the authors judge this? Do they also consider the feasibility of, for example, CDR scaling, and rule out those that exceed biomass availability constraints? Fourth, it is not acknowledged that these are techno-economic pathways that already carry the weight of an implicit but highly disputed equity principle: “least-cost”. Simply comparing the shapes of the grey curves shows that Brazil and China have extraordinarily steep pathways, but the EU and USA do not. Does it mean that it is technologically infeasible for the EU and USA to reach net negative CO₂ emissions before 2050? Or simply that it is cheaper to do so in other regions? I would strongly recommend to search for a different wording than “deepest technologically feasible”, unless the authors plan to open up all these different points.

Thank you for your thoughtful and detailed feedback. As you correctly noted, these pathways are derived from C1 category in the IPCC Sixth Assessment Report (AR6) scenario database. We have explicitly stated this in the first paragraph of in the Results subsection titled “Fair budgets against deepest available domestic reduction” (see Page 7, Line 190-193), as well as in the “Data sources” section in the Methods section of the main manuscript (see Page 17, Line 523-526).

“We take the set of emissions scenarios of the C1 category in the IPCC Sixth Assessment Report (AR6) scenario database that have country-level projections as a range of illustrative values for the deepest available domestic reduction (DADR) pathways, consistent with the target of limiting warming to 1.5°C (>50%) with no or limited overshoot.” (see Page 7, Line 190-193)

“The national DADR pathways from 2022 to 2050 are used as a comparison to our fair allocation result. They are taken from the IPCC AR6 database²¹, following a target of limiting warming to 1.5°C (>50%) with no or limited overshoot (C1 category).” (see Page 17, Line 523-526)

In response to your comments, we have revised the term to “the deepest available domestic reduction (DADR)” pathways. This change aims to more accurately reflect the characteristics of the pathways we reference. The term “available” indicates the range of pathways is based on the current available technological assumptions and expectations that are normally used by modelers in

the AR6 database. The “deepest reduction” indicates that these pathways are situated at the lower end of the emission spectrum in the AR6 database, that is the C1 category.

Regarding your concerns about the definition of “feasible,” we agree that this term could be misleading without a more comprehensive discussion of the underlying assumptions. Among pathways of AR6 database C1 category, not all of them incorporate CDR technologies. While many of these pathways are least-cost, not all are devoid of equity considerations, and those with least-cost objectives may reflect an economical consideration behind. We recognize that the inherent assumptions in the modelling process, such as the prevalence of least-cost approaches, introduce biases that could affect interpretation of these pathways. Therefore, we have avoided the use of “feasible” to avoid any potential misconceptions.

More specific comments as follows:

3. Page 2, line 48: another way to justify this further study on allocation procedures could be to set it in the context of climate damages and assistance, as for example in Fanning and Hickel 2023 (<https://www.nature.com/articles/s41893-023-01130-8>).

Thank you for providing valuable rationtile to support the significance of performing fair allocation.

We’ve added, “For example, over-emitting countries can be considered disproportionately responsible for climate-related loss and damages and should own corresponding compensation or reparation obligations²².” (see Page 2, Line 53-55).

4. Page 2, line 56: it is not clear why GWP* is specifically needed here to convert GHGs into CO2 equivalents. Actually, the rest of the manuscript does not explain this choice at all, which is surprising given that other metrics could also be selected. I suggest elaborating on this at some.

We have added more elaboration of the characteristics of GWP* in the introduction session.

“Using GWP*, however, requires normative decisions on the period over which emissions are considered as this choice materially shifts relative warming contributions¹⁶” (see Page 2, Line 64-65).

We have further described the impact of the GWP* on our findings, providing an explanation of its influence on the outcomes of our study.

“This must be caveated with a recognition of the time-dependant warming effects of short-lived climate forcers as captured by the GWP* metric underpinning our analysis. Estimated contributions to warming by these gasses under this metric are dependent on changes in flux, such that later starting years of accounting may reward countries that subsequently reduce these fluxes, without considering their initial contribution to warming when fluxes were increasing” (see Page 10, Line 261-265).

5. Page 2, line 72: figure 1 does not depict the stated elements in this sentence (“do no harm” and

“precautionary principles”)

Fig.1 does not depict “do no harm” and “precautionary principles”, as they are not equity interpretations formed by combinations of equity principles and indicators. Rather, they serve as fundamental principles to invoke the duty of fair sharing the limited amount of remaining carbon budget and motivate drawdown of budget deficit before peak, complementing the principles of equity and CBDR&RC.

We have added clarifications of the necessity of considering “do no harm” and “precautionary principles”: “Alongside and aiding their interpretation, we also consider the do-no-harm and precautionary principles described in international environmental law, which support the international environmental regime, motivating the avoidance of harm to other nations due to domestic actions and the pursual of efforts to limit climate change in spite of scientific uncertainties²³.” (see Page 3, Line 79-83).

Meanwhile, we have removed the “(Fig. 1)” reference at the end of that sentence to avoid the potential confusion.

6. Page 3, Figure 1: if one quickly scans this figure without reading the text below it, the interpretation is that all equity principles and indicators are applied to resolve a final allocation principle obtained in the bottom right box. I suggest re-drawing to avoid this interpretation. If the analysis focuses on the first/second/third ‘equity principles’, these should be upfront in the visualisation.

Thank you for the wonderful suggestions! We re-drew this figure (see Page 4, Fig.1) to evidently show the equity interpretations in the upfront: Interpretation I, Interpretation II-A, Interpretation II-B and Interpretation III. As not all equity principles and indicators are applied to each interpretation, we use vertical grey arrows to suggest which equity principle and indicator is included in a specific equity interpretation. We provide a detailed explanation in the text beneath the framework figure as well as in the title legend.

7. Page 5, line 155: how do countries follow fair allocations from 2021 if their allocation is already exhausted (as stated in the prior paragraph)?

In this study, we provide countries’ remaining fair quota in a cumulative form, and do not regulate a specific year-by-year pathway to meet their remaining fair quota. For the indicative purpose, we provide a pathway from 2022 to 2049 by attributing their remaining budget equally each year (see SI Page 2, Supplementary Fig. 1b). In 2050, all countries reach net zero.

8. Page 6, Figure 3: based on the prior comment, extending this figure axis to 2050 would show how the post-2021 ‘projection’ is arranged for each country. Beyond this, I find the mixture of indicators and equity principles confusing. The colours used to introduce each indicator in figure 1 are not carried on here, leading to a mismatch between the lines and the pies. Also the pies refer to multiple indicators, but are assigned to individual ones here. Also note that “1900”

instead of “1990” is annotated in the top left panel.

Thank you for providing valuable suggestions! Fig.3 is meant to show the depletion of countries' remaining CO₂ warming equivalent emission budgets over time, highlighting the different characteristics of this process across different nations. With this figure, we would like to show the striking fact that many very high human development countries have far exhausted their fair budget estimates even before 1990. Fig.3 focuses on past history, and we have set the axis to extend only up to 2021 to emphasize the key points, as the post-2021 allocation pathways are conceptual and may not be realistic for countries to follow.

However, we recognize the interest in providing a comprehensive view that includes both the historical situations and the post-2021 fair quotas arranged for each country. We include another figure (see SI Page 21, Supplementary Fig. 17) with axis to 2050 to show illustrate a stylized example of how country would reach their fair cumulative quotas by 2050.

Supplementary Fig. 17 The depletion of remaining CO₂ warming equivalent emission budgets over

time. This figure differs from Fig. 3, with the x-axis extended to 2050 to illustrate a stylized example of fair allocation. During the period 2022–2049, countries adhere to fair allocations with equal annual quotas. By 2050, all countries return to zero CO₂-we emissions.

We have relabeled the lines to reflect equity interpretations, ensuring that the color scheme and labels are now consistent with those in Fig. 1. Additionally, the pies have been removed to prevent any potential confusion.

The label “1900” have been corrected to “1990” in the top left panel.

9. Page 7, Figure 4: I am not sure how useful this figure is. From it, I can learn that many LDCs and developing countries have small per capita CO₂-we emissions. Is this important to know? In the right panel, I confess to having no idea. The per capita per year budget stays stable from 2022-2049: one might interpret this to mean that emissions are 0 during these years, which is probably not the intention of the figure?

This figure is a conceptual schematic of the resolution of countries’ unequal historical warming contributions by 2050 through fair warming allocations. It demonstrates the time variation of national historical CO₂-we and future quota. Fig. a shows the unevenly distributed national historical emissions per capita across country groups. Fig. b shows the process of historical inequality would be corrected through fair allocation. If countries follow the pathway from 2022 to 2049, the cumulative warming before 2050 would meet the fair quota, and all countries return to zero emissions in 2050.

We have moved Fig.4 to the SI (see SI Page 2, Supplementary Fig. 1), and added clarifications in “Supplementary Note 1: Conceptual schematic of the resolution of countries’ unequal historical warming contributions by 2050 through fair warming allocations”: “Before 2021, historical warming-equivalent CO₂ budget per capita is unevenly distributed among countries, especially between country groups. Unequal historical warming contributions have been resolved by 2050. After 2050, all countries’ fair shares turn to 0 GtCO₂-we per capita per year”.

10. Page 8, Figure 5: same problem as figure 3 that there is a mismatch of the line and pie colours with figure 1.

We have relabeled the lines of previous Fig.5 to reflect equity interpretations, ensuring that the color scheme and labels are now consistent with those in Fig. 1. Additionally, the pies have been removed to prevent any potential confusion and the legends have been revised (see Page 6, Fig.4).

11. Methods: is PRIMAP Hist-CR or Hist-TP used?

HISTCR scenario is used, where country-reported data (CRF, BUR / NIR / NC / UNFCCC DI) are prioritized over third-party data (CDIAC, FAO, Andrew, EDGAR, BP).

We’ve clarified this in the Method section as “HISTCR scenario is used, where country-reported data are prioritized over third-party data” (see Page 15, Line 443-444).

12. Supplementary figure 5: this is a very insightful figure that directly relates to discussion text in the main manuscript. I would suggest to use it instead of figure 4.

Thank you for the suggestion! We've moved Supplementary Fig. 5 to the main manuscript as the last figure (see Page 11, Fig. 6), and moved Fig.4 to the SI (see SI Page 2, Supplementary Fig. 1).

References

1. Lamboll RD, Nicholls ZR, Smith CJ, Kikstra JS, Byers E, Rogelj J. Assessing the size and uncertainty of remaining carbon budgets. *Nat Clim Chang* 2023, **13**(12): 1360-1367.
2. World Government Bond. 10Y Bond Yield. 2023 [cited 2023-02-27] Available from: <http://www.worldgovernmentbonds.com/>
3. Trading Economics. Major10Y. 2023 [cited 2023-03-01] Available from: <https://tradingeconomics.com/bonds>
4. Hannah Ritchie, Pablo Rosado, Roser M. Fossil Fuels. 2023 [cited 2023-06-20] Available from: <https://ourworldindata.org/fossil-fuels>
5. Energy Institute. Statistical Review of World Energy (2024). 2024 [cited 2024-07-31] Available from: <https://www.energyinst.org/statistical-review/>
6. Vaclav Smil. Energy Transitions: Global and National Perspectives (Second expanded and updated edition). 2017 [cited 2024-07-31] Available from: <https://vaclavsmil.com/2016/12/14/energy-transitions-global-and-national-perspectives-second-expanded-and-updated-edition/>
7. Jacks DS. From boom to bust: A typology of real commodity prices in the long run. *Cliometrica* 2019, **13**(2): 201-220.
8. ChartsBin statistics collector team 2014. Historical Crude Oil prices, 1861 to Present. 2023 [cited 2023-06-06] Available from: <http://chartsbin.com/view/oau>
9. Dudley B. BP statistical review of world energy 2018. *Energy economic, Centre for energy economics research and policy British Petroleum*, Available via <https://www.bp.com/en/global/corporate/energy-economics/statistical-review-of-world-energy/electricity.html> 2018, **5**.
10. Dudley B. BP statistical review of world energy 2016. *British Petroleum Statistical Review of World Energy, Bplc editor, Pureprint Group Limited, UK* 2019.

-
11. Schleussner C-F, Ganti G, Lejeune Q, Zhu B, Pfliederer P, Prütz R, *et al.* Overconfidence in climate overshoot. *Authorea Preprints* 2023.
 12. Prütz R, Fuss S, Lück S, Stephan L, Rogelj J. A taxonomy to map evidence on the co-benefits, challenges, and limits of carbon dioxide removal. *Communications Earth & Environment* 2024, **5**(1): 197.
 13. Dooley K, Nicholls Z, Meinshausen M. Carbon removals from nature restoration are no substitute for steep emission reductions. *One Earth* 2022, **5**(7): 812-824.
 14. Asibor JO, Clough PT, Nabavi SA, Manovic V. A country-level assessment of the deployment potential of greenhouse gas removal technologies. *J Environ Manage* 2022, **323**: 116211.
 15. Pauw WP, Castro P, Pickering J, Bhasin S. Conditional nationally determined contributions in the Paris Agreement: foothold for equity or Achilles heel? *Clim Policy* 2020, **20**(4): 468-484.
 16. Rogelj J, Schleussner C-F. Unintentional unfairness when applying new greenhouse gas emissions metrics at country level. *Environ Res Lett* 2019, **14**(11): 114039.
 17. Price PR, McMullin B, O'Dochartaigh A. Methane mitigation achievement, including agriculture, is crucial to limiting dependence on uncertain carbon dioxide removal in national carbon budgeting equitably meeting Paris goals. *2nd International Conference on Negative CO2 Emissions*. Göteborg, Sweden; 2022.
 18. Steininger KW, Williges K, Meyer LH, Maczek F, Riahi K. Sharing the effort of the European Green Deal among countries. *Nat Commun* 2022, **13**(1): 1-13.
 19. Raupach MR, Davis SJ, Peters GP, Andrew RM, Canadell JG, Ciais P, *et al.* Sharing a quota on cumulative carbon emissions. *Nat Clim Chang* 2014, **4**(10): 873-879.
 20. Pan X, Wang H, Lu X, Zheng X, Wang L, Chen W. Implications of the consumption-based accounting for future national emissions budgets. *Clim Policy* 2022, **22**(9-10): 1306-1318.
 21. Byers E, Krey V, Kriegler E, Riahi K, Schaeffer R, Kikstra J, *et al.* AR6 Scenarios Database hosted by IIASA. 2022 [cited 2023-08-25] Available from: data.ece.iiasa.ac.at/ar6/
 22. Fanning AL, Hickel J. Compensation for atmospheric appropriation. *Nature*

Sustainability 2023, **6**(9): 1077-1086.

23. Rajamani L, Jeffery L, Höhne N, Hans F, Glass A, Ganti G, *et al.* National 'fair shares' in reducing greenhouse gas emissions within the principled framework of international environmental law. *Clim Policy* 2021, **21**(8): 983-1004.

We appreciate the reviewers' valuable comments and constructive suggestions. We have carefully revised the manuscript according to these comments. Point-to-point responses are provided below. The reviewers' comments are **in black**, our responses are **in blue**, and the corresponding revisions in manuscript are **in orange**. We also include page and line numbers in the documents for editor and reviewers.

=====
Reviewer #2
=====

I thank the authors for their thorough response to my previous review. I have one outstanding concern and a few minor suggestions:

Thank you very much for taking the time to review our manuscript. We are happy to address the remaining concerns. Please find our detailed responses below.

1. In the Methods, the authors now clarify the emission data they use cover CO₂, CH₄, N₂O, and F-gases, excluding AFOLU. Presumably, this is a mistake and should be LULUCF instead of AFOLU as the manuscript implies agricultural methane emissions are included "The approach in this study ensures that historical warming contributions of countries with large non-CO₂ emissions, such as Brazil and New Zealand, are accurately accounted for."

Beforehand there is also the implication that this study includes LULUCF emissions (which is what I had assumed on my first reading) when talking about an advantage of this study over others "Typically, only CO₂ emissions excluding LULUCF are allocated". I am still unsure whether the CO₂-we analysis includes LULUCF. This becomes more unclear after seeing Figure 5a, which compares CO₂-we budgets to CO₂ excluding LULUCF and CO₂ including LULUCF. LULUCF emissions were therefore included for at least part of the analysis even if it just the comparison in Figure 5a.

Thank you for highlighting this important point of clarifying LULUCF emissions. It is correct that our intention was to exclude LULUCF emissions, not the broader AFOLU sector. We have revised the wordings to "LULUCF" (see Page 15, Line 463). In the CO₂-we analysis, LULUCF emissions are not included, as LULUCF emissions are excluded in the historical warming contribution calculations. In figure 5a, we intended to compare three allocations under the same allocation framework and equity principle: 1) our allocations of global allowable warming in CO₂-we, which cover CO₂ as well as non-CO₂ gases, 2) allocations of global CO₂ emissions excluding LULUCF, 3) allocations of global CO₂ emissions including LULUCF.

The LULUCF emissions are only included in the allocations of global CO₂ emissions including LULUCF throughout this study.

- The main text needs to be clear about whether LULUCF emissions were included or not in the CO₂-we analysis. If they are not included, it would be good to explain why not and justify their inclusion

in Fig 5a. Lines 325-331 should also be more carefully written to not imply that a strength of this study over others is that they typically do not include LULUCF.

We have now explicitly clarified in the main text (Discussion section) that LULUCF emissions were not included in the CO₂-we analysis to avoid any ambiguity.

“Nevertheless, this work still does not resolve fair shares considerations for land, land uses change and forestry (LULUCF) emissions, which remain excluded due to scientific and normative issues.” (see Page 12, Line 345-346)

This clarification has also been added to the Methods section.

“In this study, the GHG gases include CO₂, CH₄, N₂O, and F-gases, and we use the national total that contains all sectors other than LULUCF and international aviation and shipping.” (see Page 15, Line 462-463)

We have added a brief explanation in the Fig. 5a caption to clarify and justify our choice of including and excluding LULUCF emissions.

“Both the global warming and CO₂ excluding LULUCF allocations exclude LULUCF emissions, while the CO₂ including LULUCF allocation accounts for historical LULUCF emissions. This comparison illustrates how the inclusion of gases in the allocation scope impacts country rankings.” (see Page 9, Lines 253-258)

In Lines 338-339, we changed the wording to “The majority of the literature allocates fair shares excluding non-CO₂ emissions, which may overestimate remaining budgets for countries like Brazil, New Zealand and Ireland” (see Page 12, Lines 338-340)

- LULUCF data (other than the exclusion of AFOLU) is not mentioned in the methods nor are the other choices in Figure 5 (e.g. is are all sectors still included in the CO₂ comparisons) The use of LULUCF and the treatment of emissions in Fig 5a needs to clear explanation in the Methods along with a mention of the limitations of the PRIMAP LULUCF data.

We have revised the Methods section in the “Sensitivity analysis” subsection to explicitly address the exclusion and inclusion of LULUCF data. While LULUCF is excluded from the main CO₂-we analysis, LULUCF emissions are included in “CO₂ including LULUCF” scenario of Fig. 5a to provide a comparison of adjusting the allocation scope. We have also clarified whether all sectors are consistently included in these comparisons. The “CO₂ excluding LULUCF” scenario includes all sectors except LULUCF and international aviation and shipping, while “CO₂ including LULUCF” scenario include all sectors except international aviation and shipping.

“We explore the impact of adjusting the allocation scope when distributing the 1.5°C target under the Paris Agreement. Specifically, we compare our allocations of the global warming budget with two alternative scenarios: (1) allocations of global CO₂ emissions excluding LULUCF, (2) allocations of global CO₂ emissions including LULUCF (Fig. 5a). Both scenarios follow the original allocation framework and use equity Interpretation I, while adjusting the allocation to the global CO₂ budget, which is set at a remaining 500 Gt, in line with the IPCC Sixth Assessment Report, which limits warming to 1.5°C (50th percentile)¹. The first scenario accounts for CO₂ emissions excluding LULUCF for historical emissions, that is encompassing all sectors except LULUCF and international aviation

and shipping. The second scenario includes LULUCF in the CO₂ emissions. Historical LULUCF emission data is also sourced from PRIMAP.” (see Page 17, Lines 511-521)

We acknowledge the challenges with the PRIMAP LULUCF data, given that it integrates data from multiple sources with different methodologies and are not harmonized. We have now included a mention of these limitations in the revised Methods section to ensure readers interpret the allocation results with caution.

“It is important to note that the allocation result generated when PRIMAP LULUCF data is included must be treated with care as PRIMAP LULUCF data are constructed from different sources using different methodologies and are not harmonized.” (see Page 17, Line 521-523)

- As a more minor point, it would make sense to reorder Fig 5a so CO₂-we is on the left as this is the base case in the rest of the analysis.

Thank you for your suggestion regarding the illustration. We have reordered Figure 5a, placing CO₂-we on the left as recommended (see Page 9, Line 250).

The exclusion or inclusion of LULUCF has a considerable effect on the medium and high-development countries’ budgets in Fig5a (greater than whether non-CO₂ gases are included by an eye test) so this point needs more attention and clarification.

Thank you for raising this observation.

In Fig.5a, the ranking of remaining CO₂-we budget per capita is presented. A total of 117 countries experience greater changes in ranking when comparing allocations of global CO₂ emissions excluding LULUCF to those including LULUCF, as opposed to comparing allocations of global CO₂ emissions excluding LULUCF to global warming allocations. Countries with the most significant ranking changes are predominantly high and medium human development countries, including Dominica, Bolivia, Fiji, Belize, Tonga, Myanmar, Nicaragua, the Democratic Republic of the Congo, Paraguay, Mozambique, and Brazil.

These countries tend to have relatively low historical CO₂ emissions and substantial remaining CO₂-we budgets per capita, making their rankings more sensitive to variations in historical emissions. For countries like Brazil, Bolivia, and the Democratic Republic of the Congo, LULUCF emissions represent a significant portion of their total emissions, which explains why their rankings shift considerably when LULUCF is included in the allocation scope.

When analyzing the proportion of remaining CO₂-we budget, most of these countries would not experience such significant changes when comparing global CO₂ emissions excluding LULUCF to those including LULUCF, as they would when comparing allocations excluding LULUCF to global warming allocations. However, Brazil stands out as an exception to this trend. Specifically, Brazil's share is 7.0% in the CO₂ excluding LULUCF scenario, -6.7% in the CO₂ including LULUCF scenario, and 2.8% in the global warming scenario. In Brazil’s case, LULUCF emissions play a disproportionately large role across different allocation methodologies.

To address this, we have expanded the discussion to provide a more detailed explanation of how

LULUCF influences the carbon budgets per capita.

“If historical LULUCF emissions are included in the allocation scope, medium and high human development countries tend to be sensitive in the ranking of remaining per capita budgets, as LULUCF emissions constitute a significant portion of their historical CO₂ emissions (Fig. 6a).” (see Page 12, Lines 349-352)

Minor suggestions and clarifications:

2. Lines 34-38: The descriptions of the Paris pledges could be explained more clearly. Using “many governments” does not imply the “near-global coverage” of pledges. It would also be good to distinguish between NDCs and LTSs as there is not a near-global coverage of LTSs.

Thank you for pointing out this distinction in national pledges. We have revised the text to distinguish between NDCs and LTSs. We now emphasize that nearly all governments have submitted NDCs and around one-third have submitted long-term strategies.

“Nearly all governments have submitted Nationally Determined Contributions (NDCs) under the Paris Agreement² to contribute to the agreement’s aims to hold warming well-below 2°C while pursuing to limit it to 1.5°C³, and around one-third have submitted long-term strategies⁴. However, the broad coverage of NDCs does not guarantee that normative concerns are met⁵ as countries seldom account for their full historical responsibility for causing climate change, or their relative capability to meet deep emissions reductions⁵.” (see Page 2, Lines 34-40)

3. I still find the heading “Results” before “Interpretation of equity principles” strange as this reads more like a methods section. It would make more sense to me to have “Results” before the “Fair national CO₂ warming equivalents” section. This may be a point for the Editor’s input as the previous response referred to the Nature Communications guidelines.

Thank you for your feedback on the structure of the manuscript. In the last round of revision, we left the section sequence unchanged for consistency with the journal’s formatting guidance, but we have revised part of the introductory text of the “Results” section to better clarify the transition from the introduction section to the interpretation of principles and quantitative findings. We are open to any further editorial input on this matter if needed.

4. I find the new inclusion of different equity interpretation names (e.g. Interpretation I) helpful. It would be nice to revise the “calculation of the allocation procedure” section of the Methods to also include these names for consistency and perhaps add subheadings to help guide readers when reading this section.

Thank you for your positive feedback of including the equity interpretation names. We have now included these names in the “Calculation of the allocation procedure” section of the Methods to clarify how these interpretations are derived in our allocation procedure.

“Fair share Interpretations I, II-A, II-B, and III are formed through these steps. All interpretations

include Steps 1, 2, and 4, while Interpretations II-A, II-B, and III also include Step 3.” (see Page 15, Lines 401-402)

“In the third step, we interpret either the ability-to-pay principle for Interpretation II or the beneficiary-pays principle for Interpretation III. We translate the principles with measurable indicators, including the 10-year bond yield (Interpretation II-A), GDP per capita (Interpretation II-B), and historical fossil fuel sales (Interpretation III).” (see Page 15, Lines 422-425)

We have divided this section in to four parts, consistent with the allocation procedure outlined in Fig. 1, and have added subheadings: “Step 1: determining the allocation target”; “Step 2: allocating initial national allowable warming budgets”; “Step 3: adjusting national allowable warming budgets”; “Step 4: calculating remaining national allowable warming budgets”.

5. Line 93: The justification for 1990 being used (i.e. excusable ignorance) is mentioned later in the manuscript but not here where 1990 is first mentioned. I would move the justification here to help explain this choice on its first mention.

Thank you for this nice suggestion. We have moved the relevant justification from around line 492 to line 93 where the choice of starting year first occur (see Page 3, Lines 93-98).

5. Fig 1: What does “before peak” mean? Is this before peak warming? There is potential for misinterpretation (e.g. before peak emissions) given the context.

Thank you for reminding the potential misinterpretation. In this context, “before peak” indeed means before peak warming. We’ve clarified the wording in Fig.1 to clarify this.

Fig. 1 Analysis framework linking equity principles, indicators, and the allocation procedure to

estimate fair global warming contributions. Each equity principle and the corresponding indicators are marked in different colours: pink stands for the principle of equality, light blue for ability to pay, light green for beneficiary pays and yellow for polluter pays. Interpretations are marked in different colours: dark blue stands for Interpretation I, orange for Interpretation II-A, dark green for Interpretation II-B and red for Interpretation III. The horizontal coloured arrows illustrate the incorporation of equity principles and indicators into the allocation procedure. Under the “Indicators” column, plus (or minus) signs in brackets indicate the indicators are directly (or inversely) proportional to the remaining national allowable warming budgets. Under the “Interpretation” columns, the vertical grey arrows suggest whether the corresponding indicator is included in this step.

6. Lines 127-128: better to add something along the lines of “as defined by UNDP HDI ranking”.

Thank you for this suggestion. We have added the clarification as suggested.

“We explore the quantitative results through the lens of four country groups: very high, high, medium, and low human development countries, as defined by the United Nations Development Programme (UNDP) Human Development Index (HDI) ranking.” (see Page 4, Line 133-135)

7. Lines 166-174: In response to point 13 of my previous review, the authors state that they no longer report ranges from the remaining budgets. However, ranges are still reported in lines 166-174 in the revised manuscript. These should either be changed to single values or the uncertainty in the ranges clarified.

Thank you for pointing this out. We have now removed all ranges in the manuscript, and present the result under equity Interpretation I instead (see Page 6, Lines 170-179). The result under the other interpretations or cases could be found in the supplementary datasets.

8. Line 275-276: South Korea is not really a good example of a country relying “primarily on the tertiary sector” as they have a strong industrial sector. It is also a bit of an exaggerated statement for Japan and Switzerland as about a quarter of their GDPs are from industry.

Thank you for highlighting this. Our intention of the statement was to emphasize certain economies that are driven by sectors less dependent on the fossil fuels. These economies may argue that they derive fewer direct economic benefits from emissions compared to those more reliant on fossil fuel extraction and heavy industry.

To address your comment, we have adjusted our examples and revised the text to more accurately reflect this information.

“Very high human development countries like Switzerland and Finland, whose economies have historically and currently been primarily driven by high-value industrial sectors and the tertiary sector may emphasize that they have limited extraction of fossil energy and thus limited benefit from emissions.” (see Page 10, Lines 284-287)

9. Finally, the different equity interpretations have a considerable focus in the paper but do not have much influence on the overall rankings of countries (the coverage of gases and sectors seems much more significant). Perhaps the authors could comment a bit more on their interpretation of this. Do they feel that the differences arising from different equity interpretations are small enough that the most simple Interpretation I is sufficient?

The different interpretations reflect the diverse moral perspectives on fair sharing, which are frequently debated in climate negotiations and extensively discussed in the literature (more so than parameters like gas coverage or sectoral scope). Our aim in presenting multiple interpretations was to recognize this diversity and explore how different perspectives might influence national allocations. Interpretations turn out to be the second most significant influential factor. While it is true that these equity interpretations may not have the most major impact on the numerical rankings, their inclusion provides valuable insights for stakeholders with differing normative views. We have used Interpretation I as the central case because it is both intuitive and widely recognized. In the main text, we focus on this interpretation to ensure simplicity and clarity, while we present the variations caused by different equity interpretations in Fig. 5b and Supplementary Information to demonstrate that these differences do not alter our overall qualitative conclusions. While Interpretation I provides a baseline for our allocation results, it is not intended as the definitive solution but rather as a reference point.

We have added more comments for this as suggested.

“In total, we develop three interpretations of how these principles can be combined, to reflect the diverse moral perspectives frequently debated in climate negotiations.” (see Page 3, Lines 85-87)

“While Interpretation I serves as a central case for presenting our allocation results, it is not intended as a definitive solution but rather as an illustrative reference point.” (see Page 9, Lines 222-224)

=====

Reviewer #3

=====

The authors have adequately addressed my comments in the first round of review and I would be happy to now see this piece published.

Thank you very much for your positive feedback and for taking the time to review our manuscript. We really appreciate your comments and are glad to hear that our revisions have adequately addressed your concerns.

References

1. Masson-Delmotte V, Zhai P, Pirani A, Connors SL, Péan C, Berger S, *et al.* Climate change 2021: the physical science basis. *Contribution of working group I to the sixth assessment report of the intergovernmental panel on climate change 2021*, **2(1)**: 2391.

-
2. UNFCCC. Adoption of the Paris Agreement. Paris, France: United Nations Framework Convention on Climate Change; 2015. Report No.: No. FCCC/CP/2015/L. 9/Rev. 1, 21932
 3. United Nations Climate Change. Long-term strategies portal. 2024 [cited 2024-10-16] Available from: <https://unfccc.int/process/the-paris-agreement/long-term-strategies>
 4. Winkler H, Höhne N, Cunliffe G, Kuramochi T, April A, de Villafranca Casas MJ. Countries start to explain how their climate contributions are fair: more rigour needed. *International environmental agreements: Politics, law and economics* 2018, **18**: 99-115.